chemical biology

terpyridine, fluorescence, anti-microbial activity, molecular docking, density functional theory, structure–activity relationship

**Authors for correspondence:**
Ehsan Ullah Mughal
e-mail: ehsan.ullah@uog.edu.pk
Masoud Mirzaei
e-mail: mirzaeesh@um.ac.ir

This article has been edited by the Royal Society of Chemistry, including the commissioning, peer review process and editorial aspects up to the point of acceptance.

# Terpyridine-metal complexes: effects of different substituents on their physico-chemical properties and density functional theory studies

Ehsan Ullah Mughal[1], Masoud Mirzaei[2], Amina Sadiq[3], Sana Fatima[1], Ayesha Naseem[1], Nafeesa Naeem[1], Nighat Fatima[4], Samia Kausar[1], Ataf Ali Altaf[1,5], Muhammad Naveed Zafar[6] and Bilal Ahmad Khan[7]

[1]Department of Chemistry, University of Gujarat, Gujarat 50700, Pakistan
[2]Department of Chemistry, Faculty of Science, Ferdowsi University of Mashhad, PO Box 9177948974, Mashhad, Iran
[3]Department of Chemistry, Government College Women University, Sialkot 51300, Pakistan
[4]Department of Pharmacy, COMSATS University Islamabad, Abbottabad Campus, 22060, Pakistan
[5]Department of Chemistry, University of Okara, Okara 56300, Pakistan
[6]Department of Chemistry, Quaid-i-Azam University, Islamabad 45320, Pakistan
[7]Department of Chemistry, University of Azad Jammu and Kashmir, Muzaffarabad, Pakistan

MM, 0000-0002-7256-4601; AAA, 0000-0001-8018-5890

A series of different substituted terpyridine (tpy)-based ligands have been synthesized by Kröhnke method. Their binding behaviour was evaluated by complexing them with Co(II), Fe(II) and Zn(II) ions, which resulted in interesting coordination compounds with formulae, $[Zn(tpy)_2]PF_6$, $[Co(tpy)_2](PF_6)_2$, $[Fe(tpy)_2](PF_6)_2$ and interesting spectroscopic properties. Their absorption and emission behaviours in dilute solutions were investigated in order to explain structure–property associations and demonstrate the impact of different aryl substituents on the terpyridine scaffold as well as the role of the metal on the complexes. Photo-luminescence analysis of the complexes in acetonitrile solution revealed a transition from hypsochromic to bathochromic shift. All the compounds displayed remarkable photo-luminescent properties and various maximum emission peaks owing to the different nature of the functional groups. Furthermore, the anti-microbial potential of ligands and complexes was evaluated with docking analyses carried out to investigate the binding affinity of terpyridine-based ligands along with corresponding proteins (shikimate dehydrogenase

and penicillin-binding protein) binding sites. To obtain further insight into molecular orbital distributions and spectroscopic properties, density functional theory calculations were performed for representative complexes. The photophysical activity and interactions between chromophore structure and properties were both investigated experimentally as well as theoretically.

## 1. Introduction

2,2′:6′,2″-Terpyridine is a tridentate ligand which contains three coordination sites belonging to *N*-heteroaromatic rings (figure 1) and thus constitutes a significant class of aromatic heterocyclic compounds. Owing to their strong chelating tendency, these ligands can form stable complexes with several different main groups and transition metal ions. In the past, terpyridine motifs and their complexes have attracted the increasing attention of materials chemists due to their applications in several fields, for instance photovoltaic devices, DNA binders, sensors, photosensitizers, molecular chemistry, medicinal chemistry and metal-organic framework (MOF) construction [1–3].

Moreover, their complexes with transition metals, in particular, can lead to unique photo-luminescence, catalysis, sensor properties and quite promising tumour-inhibiting activities [4–19]. Due to environmental and economic considerations, increasing attention has been paid to the development of such compounds for use in many fields, especially in medicinal and material chemistry. Constable's group has thoroughly studied the synthesis of terpyridine derivatives, and a wide variety of substituted terpyridine ligands were prepared in high yields through the Kröhnke reaction by condensation of 2-acetylpyridine with various substituted aryl aldehydes followed by oxidation in simple and effective processes [20–25]. These ligands show unique coordinative capabilities towards transition elements and have received much attention recently because of their varied/extensive uses in diverse research fields extending from therapeutic uses (such as anti-cancer and DNA intercalation) to material sciences (photovoltaics, sensitizers) and catalysis [26]. Supramolecular chemistry and material science are another field where 2,2′:6′,2″-terpyridines and their derived complexes are widespread and used as significant organic ligands [27–30]. The features of the metal-containing assemblies depend on the electronic impact of both the terpyridine unit and its substituents as well as the metal ion [31]. These compounds provide an opportunity to probe the effects of the ligand, the central metal ion and the coordination geometry on the binding properties. 2,2′:6′,2″-Terpyridine and its structural derivatives with prolific coordination chemistry were broadly considered for their strong binding affinity for a variety of transition metal ions resulting in various metal supramolecular architectures with fascinating photophysical and redox characteristics [32–36]. Hofmeier *et al.* [34] and Alcock *et al.* [12] have synthesized ligands based on terpyridine and developed radiative metallo-supramolecular coordination compounds accumulated through π–π fibre optics in which violet-blue emissions are ascribed to π–π fibre optic ligand transfer. It has been shown that a donor-acceptor system is an efficient approach for adapting the optical properties of organic–inorganic hybrid materials [37,38]. Moreover, transition metal complexes give many advantages, for example, long-lived fluorescent excited states and photochemical strengths [39]. Furthermore, the core-functionalization of terpyridine units with electron-donating/withdrawing substituents has not been developed as much as possible fluorescent probes. The studies indicated that electron-accepting and -donating substituents may be used to modify the photophysical and oxidation-reduction properties of free terpyridines and their metal complexes [40]. Therefore, to obtain more knowledge of the structure–activity correlation (absorption and emission properties), a great variety of substituents having different electron-releasing or -accepting behaviour were introduced on the *p*-position of the terpyridine rings.

Furthermore, there has been recent interest in the terpyridine moiety, not only because of its exciting molecular topologies in the design and synthesis of polymers for coordination but also because they have strong anti-microbial activity, which could provide an important point of reference for more effective anti-microbial drug design [41–48]. This increased lipophilicity can lead to a breakdown of the cell permeability barrier, thereby blocking normal cell processes [49–52]. Previous studies [53,54] showed that bidentate or tridentate ligands with greater lipophilicity had higher anti-microbial activity than monodentate ligands and, as stated by the principle of similarity and inter-miscibility, two different types of ligands in complexes had stronger anti-bacterial activity than a single ligand. It would be motivated to discover the structure–property relationship between complex structures and anti-microbial properties, which can offer theoretical directions for designing and synthesizing complexes with useful biological activities [55–64].

**Figure 1.** Structural representation of terpyridine framework.

Based on the aforementioned considerations, we report the design, synthesis and characterization of variously functionalized terpyridine ligands and their Zn(II), Co(II), and Fe(II) complexes. Thorough photophysical and computational studies of the new complexes provide an insight into their electronic structure, which provides a significant extension of newly published terpyridine-transition metal complexes. In addition, we have established a detailed structure–activity relationship (SAR) between these architectures and their biological and spectrophotometric properties.

# 2. Materials and methods

All chemicals were bought from Merck and Sigma-Aldrich and used as received. Melting points have been recorded with an electrothermal device and are uncorrected. The FTIR spectra have been obtained on a Bio-Rad spectrophotometer and the infrared values are listed in $\bar{\upsilon}$ units. The UV-Vis spectra were recorded in chloroform ($CHCl_3$) and acetonitrile ($CH_3CN$) solutions, respectively, on a Jasco UV-Vis V-660 instrument using a QUARTZ cell. The luminescence spectra were obtained using a Shimadzu 8101AFT-IR instrument. Thin-layer chromatography (TLC) was conducted on silica gel TLC plates purchased from Merck and chromatograms were viewed with a UV-lamp at 254 and 365 nm.

## 2.1. General procedure for the synthesis of 4′-substituted terpyridine ligands and complexes

### 2.1.1. Synthesis of ligand (4′-aryl-substituted 2,2′:6′,2″-terpyridine) [65]

To a mixture of 2-acetylpyridine (2.43 g, 20.0 mmol) in methanol (20 ml), a substituted aryl aldehyde (10.0 mmol) was added, followed by potassium hydroxide (KOH) pellets (1.54 g, 24 mmol) and 35% aqueous ammonia solution (40.0 ml). The reaction mixture was refluxed for 4–6 h. After the reaction was completed (determined by TLC monitoring), the solvent was removed under vacuum and the precipitates were filtered, washed with plenty of distilled $H_2O$ to remove the excess base and finally with ice-cold EtOH until the washings were neutral. The residue was dried in the open air and recrystallized from ethanol.

### 2.1.2. Synthesis of terpyridine metal complexes (**C₁–C₂₇**) [66]

A hot methanolic solution (20.0 ml) of metal salt (0.5 mmol) was introduced dropwise to a $CH_2Cl_2$ solution (20.0 ml) of the substituted terpyridine ligand (0.28 g, 1.0 mmol) under continuous stirring. The colour of the reaction mixture instantly changed and the reaction mixture was stirred for approximately 2 h at room temperature. Upon addition of an excess of $NH_4PF_6$ precipitation occurred. The precipitates were filtered, washed with ice-cold MeOH (5.0 ml) and $(C_2H_5)_2O$ (15.0 ml) to obtain a pure complex. Recrystallization from acetonitrile or methanol or a mixture of both yielded the analytically purified complex.

## 2.2. Photo-luminescence studies

RF-6000 Spectrofluorophotometer was used to produce an emission spectrum for ligands and their complexes. The emission spectra of the synthesized ligand in chloroform and of complexes in acetonitrile were obtained by using a fluorescence spectrophotometer. The emission spectra were checked by keeping the excitation value of the synthesized compounds at 325 nm [40,67–72]. The fluorescence spectra have been observed at 335 nm for ligands (**L₁–L₉**) and 520 nm for complexes (**C₁–C₂₇**).

## 2.3. Molecular docking studies

### 2.3.1. Accession of targets proteins

Chemical structures of the terpyridines were developed with ChemBio Draw which also produced their MOL format files. The shikimate dehydrogenase (PDB ID: 3DON) and penicillin-binding protein (PDB ID: 1VQQ) three-dimensional crystal structures have been accessed and downloaded from the protein data bank database (PDB) [http://www.rcsb.org/pdb/home/home.do] [71].

### 2.3.2. Analysis of target active binding sites

The active sites of the target protein were examined using the molecular operating environment (MOE) software. An active site was distinct from the ligand coordinates in the original protein sites [40,72–75].

### 2.3.3. Docking analysis

A theoretical ligand-target docking method has been used to classify structural complexes of the shikimate dehydrogenase crystal structure and penicillin-target protein with ligand molecules to recognize the molecular basis of the specificity of these protein targets. Eventually, the MOE program performs docking. The energy of these derivatives to interact with the protein targets is given 'grid level'.

## 2.4. Density functional theory studies

Computational calculations were performed by using DFT-B3LYP*(DZ) in the Amsterdam Density Functional (ADF) modelling suite [76]. The ground states geometries of metal complexes (**C$_{13}$, C$_{14}$, C$_{15}$, C$_{22}$, C$_{23}$ and C$_{24}$**) were optimized using the B3LYP* hybrid functional and double zeta (DZ) basis sets.

## 2.5. Anti-microbial studies

### 2.5.1. Anti-bacterial activity

All the new compounds were assessed using a disc-diffusion method for the *in vitro* anti-bacterial activity against Gram-negative *Escherichia coli* (ATCC 25922) and *Pseudomonas aeruginosa* (ATCC 9721) and Gram-positive *Staphylococcus aureus* (ATCC 6538) bacterial strains. Cefixime was used as a positive standard and dimethyl sulfoxide (DMSO) as a negative control. Various sample solutions were prepared by dissolving 4.0 mg of each sample in 1.0 ml of DMSO. The lawn was developed on nutrient agar plates using bacterial strains of equal turbidity that is accomplished using 0.5% of McFarland's solution. The well depth was 8 mm and they were all made at the correct distances from each other. The sample and standard are poured into their respective tubes. The sample quantity used was 80 µl in a well together with the two controls. The prepared dishes were incubated for 24 h at 37°C, and the findings were reported as the average diameter of zone of inhibition (ZOI) of bacterial development around the discs in millimetres for each compound [75,77].

## 2.6. Anti-fungal activity

The new compounds have also been tested for anti-fungal activity, using the process of well diffusion [75,77]. Two fungal strains were used for these experiments: one *Candida albicans* (ATCC 9002) and a second *Candida parapsilosis* (ATCC 22019). To relate the activities of target compounds, clotrimazole (1 mg ml$^{-1}$ in DMSO) was used as a standard. The inhibition zone was calculated to obtain quantitative activity information. The sample solutions were prepared by adding 4.0 mg of sample in 1.0 ml solvent (DMSO). Using a sterile glass rod, a spore suspension aliquot (1108 spores ml$^{-1}$) was spread evenly over Sabouraud dextrose agar (SDA). Eight-millimetre diameter wells were built at suitable ranges. An 80 µl aliquot of sample was introduced into each well using a micropipette and the loaded plates were incubated at 28°C for 24–28 h. Anti-fungal behaviour was measured as the circular diameter of the zone of inhibition in millimetres.

# 3. Results and discussion

## 3.1. Chemistry

The synthesis of terpyridine ligands (**L$_1$–L$_9$**) was carried out by Kröhnke method [65]. Our preliminary study focused on the synthesis of different 6,6″ symmetrically substituted 4′-aryl-2,2′:6′,2″-terpyridine

substrates. As outlined in scheme 1, the aromatic aldehydes were condensed with 2-acetylpyridine in the presence of methanolic-potassium hydroxide and aqueous ammonia solution to furnish terpyridine ligands ($L_1$–$L_9$) in moderate to excellent yields. Purification of the ligands was achieved by recrystallization from ethanol. The chemical structures of all the ligands were corroborated by UV-Vis, FTIR and NMR spectroscopies. Subsequently, these ligands were coordinated with the transition metal cations $Fe^{2+}$, $Co^{2+}$, $Zn^{2+}$ to produce the desired complexes ($C_1$–$C_{27}$) in very good yields. These complexes were prepared by the reaction of the terpyridine ligands with the metal salts in a 2 : 1 molar ratio in dichloromethane (DCM) and purified by washing with ethanol. The ligands ($L_1$–$L_9$) and their complexes ($C_1$–$C_{27}$) are stable at ambient temperature and readily soluble in chloroform and/or acetonitrile. Characterization of the complexes was accomplished from their FTIR and $^1$H NMR and $^{13}$C NMR spectra. The FTIR spectra were acquired over the 4000–400 cm$^{-1}$ range. For ligands ($L_1$–$L_9$), characteristic bands in the regions 3013–3062 and 1597–1660 cm$^{-1}$ are assigned to $v_{C-H}$ and $v_{C=N}$ stretching vibrations, respectively. The bands in the region 1443–1530 cm$^{-1}$ are due to C=C stretching vibration, and the band in the range 1240–1358 cm$^{-1}$ is assigned to the C-N group, but in the case of metal complexes ($C_1$–$C_{27}$), the C-H group disappeared. Moreover, the presence of a strong absorption peak in the 1587–1615 cm$^{-1}$ range is attributed to the azomethine moiety indicating the formation of a -CH=N bond. However, on complexation, the characteristic $v$(C=N) band was shifted towards the lower frequency suggesting coordination of the metal with the terpyridine moiety in the ligand framework.

The electronic absorption spectra of the compounds were recorded in the 200–800 nm range in chloroform and acetonitrile solutions. The compounds show absorption bands in the 234–350 nm range which may be assigned to $\pi$-$\pi^*$ energy transitions of the C=N and phenyl moieties. Finally, coordination of the new ligands with various metal cations and the molar masses of the resulting complexes were confirmed by MALDI mass spectrometers. Spectroscopic data are in good agreement with the proposed structures of all ligands as well as complexes. Unfortunately, various attempts to grow a single crystal of the complexes for X-ray diffraction analysis proved unsuccessful.

The physical and spectroscopic data of all the new ligands and complexes are given below:

## 3.2. 4′-(4-Methylphenyl)-2,2′:6′,2″-terpyridine ($L_1$) [68]

Colourless crystalline solid; Yield: 62%; m.p. 162–167°C; UV $\lambda_{max}$ (CHCl$_3$) = 325 nm; $\lambda_{em}$ (CHCl$_3$): 357, 731 nm; FTIR (cm$^{-1}$): 3013 (C-H), 1601 (C=N), 1464 (C=C), 1311 (C-N), 1257 (C-H), 1066 (C-H); $^1$H NMR (300 MHz, CDCl$_3$) $\delta$ 8.73–8.70 (m, 4H, Ar-H), 8.65 (d, $J$ = 9.0 Hz, 2H, Ar-H), 7.86–7.84 (m, 4H, Ar-H), 7.84 (d, $J$ = 9.0 Hz, 2H, Ar-H), 7.35–7.30 (m, 2H, Ar-H), 2.42 (s, 3H, -CH$_3$); $^{13}$C NMR (75 MHz, CDCl$_3$) $\delta$ 156.4, 156.0, 151.2, 150.0, 137.0, 134.3, 130.0, 128.2, 124.0, 121.5, 118.8, 21.3, other carbons are isochronous.

## 3.3. N,N-dimethyl-4-(2,2′:6′,2″-terpyridin-4′-yl)aniline ($L_2$) [78]

Green amorphous solid; Yield: 71%; m.p. 180–195°C; UV $\lambda_{max}$ (CHCl$_3$) = 293 nm; $\lambda_{em}$ (CHCl$_3$): 344, 443 nm; FTIR (cm$^{-1}$): 2809 (C-H), 1595 (C=N), 1464 (C=C), 1358 (C-N), 1190 (C-H), 1045 (C-H); $^1$H NMR (300 MHz, CD$_3$CN) $\delta$ 8.70 (d, $J$ = 9.0 Hz, 2H, Ar-H), 8.52–8.45 (m, 4H, Ar-H), 8.31 (d, $J$ = 9.0 Hz, 2H, Ar-H), 7.93 (s, 2H, Ar-H), 7.93–7.82 (m, 1H, Ar-H), 7.50–7.48 (m, 1H, Ar-H), 7.24–7.19 (m, 1H, Ar-H), 7.08–7.03 (m, 1H, Ar-H), 3.46 (s, 6H, -CH$_3$); $^{13}$C NMR (CD$_3$CN, 75 MHz) $\delta$ 160.0, 159.4, 157.3, 156.3, 154.8, 153.5, 152.9, 138.6, 138.5, 135.2, 128.3, 127.7, 125.0, 124.7, 124.3, 107.4, 40.8, other carbons are isochronous.

## 3.4. N,N-diphenyl-4-(2,2′:6′,2″-terpyridin-4′-yl)aniline ($L_3$)

Yellow amorphous solid; Yield: 75%; m.p. 160–162°C; UV $\lambda_{max}$ (CHCl$_3$) = 234 nm; $\lambda_{em}$ (CHCl$_3$): 335, 576 nm; FTIR (cm$^{-1}$): 3054 (C-H), 1660 (C=N), 1565 (C=C), 1271 (C-N), 1120 (C-H); $^1$H NMR (300 MHz, Acetone-$d_6$) $\delta$ 9.03 (bs, 1H, Ar-H), 8.87 (d, $J$ = 9.0 Hz, 1H, Ar-H), 8.62–8.56 (m, 1H, Ar-H), 8.37 (d, $J$ = 9.0 Hz, 2H, Ar-H), 8.11–7.97 (m, 3H, Ar-H), 7.90–7.85 (m, 1H, Ar-H), 7.72–7.67 (m, 2H, Ar-H), 7.55–7.47 (m, 2H, Ar-H), 7.27 (t, $J$ = 9.0 Hz, 2H, Ar-H), 7.14–7.01 (m, 4H, Ar-H), 6.88 (d, $J$ = 9.0 Hz, 2H, Ar-H); $^{13}$C NMR (75 MHz, Acetone-$d_6$) $\delta$ 162.7, 156.7, 156.2, 147.4, 147.1, 138.1, 130.7, 130.3, 129.7, 129.6, 129.5, 128.4, 126.8, 126.3, 125.5, 125.4, 124.6, 122.4, 121.1, 118.4, other carbons are isochronous.

**Scheme 1.** Synthetic routes to the ligands (**L₁–L₉**) and their complexes (**C₁–C₂₇**).

## 3.5. 4′-(4-Methoxyphenyl)-2,2′:6′,2′′-terpyridine (**L₄**) [79,80]

Dark-brown amorphous solid; Yield: 62%; m.p. 162–164°C; UV $\lambda_{max}$ (CHCl₃) = 248 nm; $\lambda_{em}$ (CHCl₃): 371, 746, 855 nm; FTIR (cm⁻¹): 3024 (C-H), 1689 (C=N), 1466 (C=C), 1305 (C-N), 1236 (C-H), 1150 (C-H); ¹H NMR (300 MHz, CDCl₃) δ 8.65–8.54 (m, 4H, Ar-H), 8.50 (d, J = 9.0 Hz, 2H, Ar-H), 7.80–7.82 (m, 4H, Ar-H), 7.83 (d, J = 9.0 Hz, 2H, Ar-H), 7.32–7.29 (m, 2H, Ar-H), 3.9 (s, 3H, -OCH₃); ¹³C NMR (75 MHz, CDCl₃) δ 156.4, 156.0, 151.2, 150.0, 137.0, 134.3, 130.0, 128.2, 124.0, 121.5, 118.8, 54.2, other carbons are isochronous.

## 3.6. 4′-(3-Trifluoromethylphenyl)-2,2′:6′,2′′-terpyridine (**L₅**)

Off-white crystalline solid; Yield: 70%; m.p. 135–136°C; UV $\lambda_{max}$ (CHCl₃) = 234 nm; $\lambda_{em}$ (CHCl₃): 359, 727, 861 nm; FTIR (cm⁻¹): 3356 (C-H), 1604 (C=N), 1454 (C=C), 1281 (C-N), 1152 (C-H); ¹H NMR (300 MHz, CDCl₃) δ 8.31–8.28 (m, 4H, Ar-H), 8.24–8.21 (m, 2H, Ar-H), 7.69 (s, 1H, Ar-H), 7.64 (d, J = 9.0 Hz, 1H, Ar-H), 7.47 (ddd, J = 3.0, 6.0, 9.0 Hz, 2H, Ar-H), 7.29 (d, J = 9.0 Hz, 1H, Ar-H), 7.22–7.17 (m, 1H, Ar-H),), 6.94–6.90 (m, 2H, Ar-H); ¹³C NMR (75 MHz, CDCl₃) δ 156.4, 156.0, 155.6, 149.1, 148.8, 139.0, 137.0, 136.9, 130.7, 130.4, 130.1, 129.3, 126.0, 125.6, 125.4, 124.1, 124.0, 121.4, 121.2, 118.5, 77.4,

## 3.7. 4′-(2-Trifluoromethylphenyl)-2,2′:6′,2′′-terpyridine (**L₆**) [81]

Light-yellow crystalline solid; Yield: 64%; m.p. 148–150°C; UV $\lambda_{max}$ (CHCl₃) = 240 nm; $\lambda_{em}$ (CHCl₃): 345, 70, 80 nm; FTIR (cm⁻¹): 3362 (C-H), 1655 (C=N), 1493 (C=C), 1311 (C-N), 1280 (C-H), 1150 (C-H); ¹H NMR (300 MHz, CDCl₃): δ 8.72–8.70 (m, 4H, Ar-H), 8.54 (s, 2H, Ar-H), 7.84–7.77 (m, 2H, Ar-H), 7.54–7.49 (ddd, J = 3.0, 6.0, 9.0 Hz, 2H, Ar-H), 7.35 (d, J = 9.0 Hz, 2H, Ar-H), 7.25–7.20 (m, 2H, Ar-H); ¹³C NMR (75 MHz, CDCl₃) δ 156.3, 156.1, 155.5, 150.0, 149.0, 139.0, 137.1, 137.0, 130.6, 131.0, 130.0, 129.2, 126.1, 125.5, 125.4, 124.1, 123.8, 121.2, 121.0, 118.4, 77.3.

## 3.8. 4′-(4-Trifluoromethylphenyl)-2,2′:6′,2′′-terpyridine (**L₇**)

Off-white amorphous solid; Yield 68%; m.p. 129–131°C; UV $\lambda_{max}$ (CHCl₃) = 234 nm; $\lambda_{em}$ (CHCl₃): 345, 723, 852 nm; FTIR (cm⁻¹): 3361 (C-H), 1619 (C=N), 1469 (C=C), 1326 (C-N), 1261 (C-H), 1143 (C-H); ¹H NMR (300 MHz, CDCl₃) δ 8.66–8.63 (m, 4H, Ar-H), 8.61–8.58 (m, 2H, Ar-H), 7.93 (d, J = 9.0 Hz,

| compound no. | structures |
|:---:|:---:|
| **L₁** |  |
| **L₂** |  |
| **L₃** |  |
| **L₄** |  |
| **L₅** |  |
| **L₆** |  |

**Scheme 1.** (*Continued.*)

2H, Ar-H), 7.83 (ddd, $J$ = 3.0, 6.0, 9.0 Hz, 2H, Ar-H), 7.67 (d, $J$ = 9.0 Hz, 2H, Ar-H),), 7.30–7.26 (m, 2H, Ar-H); $^{13}$C NMR (75 MHz, CDCl₃) $\delta$ 156.3 156.1, 149.0, 142.0, 139.0, 137.1, 127.3, 126.0, 126.0, 124.2, 124.0, 123.2, 121.4, 119.5, 119.1, 77.4, other carbons are isochronous.

| L7 |  |
| L8 |  |
| L9 |  |
| C1 |  |
| C2 |  |
| C3 |  |
| C4 |  |

**Scheme 1.** (*Continued.*)

| | |
|---|---|
| **C$_5$** |  .2PF$_6$ |
| **C$_6$** |  .2PF$_6$ |
| **C$_7$** |  |
| **C$_8$** |  .2PF$_6$ |
| **C$_9$** |  .2PF$_6$ |
| **C$_{10}$** |  |

**Scheme 1.** (*Continued.*)

## 3.9. 4′-(3-Nitrophenyl)-2,2′:6′,2″-terpyridine (**L$_8$**) [82]

Brown amorphous solid; Yield 75%; m.p. 151–153°C; UV $\lambda_{max}$ (CHCl$_3$) = 253 nm; $\lambda_{em}$ (CHCl$_3$): 361, 723, 849 nm; FTIR (cm$^{-1}$): 3402 (C-H), 1613 (C=N), 1469 (C=C), 1346 (C-N), 834 (C-H), 787 (C-H); $^1$H NMR (300 MHz, CDCl$_3$) $\delta$ 8.75 (d, $J$ = 9.0 Hz, 1H, Ar-H), 8.69–8.65 (m, 4H, Ar-H), 8.62 (d, $J$ = 9.0 Hz, 2H, Ar-H), 8.26–8.22 (m, 1H, Ar-H), 8.16–8.12 (m, 1H, Ar-H), 8.85–8.79 (m, 2H, Ar-H), 7.65–7.58 (m, 1H, Ar-H), 7.33–7.26 (m, 2H, Ar-H); $^{13}$C NMR (75 MHz, CDCl$_3$) $\delta$ 156.4, 156.1, 155.9,

| | |
|---|---|
| **C₁₁** |  |
| **C₁₂** |  |
| **C₁₃** |  |
| **C₁₄** |  |
| **C₁₅** |  |
| **C₁₆** |  |

**Scheme 1.** (*Continued.*)

154.9, 148.9, 147.3, 140.5, 136.9, 133.2, 129.9, 125.3, 124.0, 123.9, 123.8, 123.0, 122.8, 122.3, 121.8, 120.7, 119.1, 118.6.

## 3.10. 4′-(4-Nitrophenyl)-2,2′:6′,2″-terpyridine (L₉)

Reddish-brown amorphous solid; Yield 70%; m.p. 149–151°C; UV $\lambda_{max}$ (CHCl₃) = 247 nm; $\lambda_{em}$ (CHCl₃): 403, 720, 851 nm; FTIR (cm$^{-1}$): 3042 (C-H), 1582 (C=N), 1514 (C=C), 1265 (C-N), 1127 (C-H); $^1$H NMR (300 MHz, CDCl₃) $\delta$ 8.74–8.68 (m, 4H, Ar-H), 8.65–8.60 (m, 2H, Ar-H), 8.25 (d, $J$ = 9.0 Hz, 2H, Ar-H),

| | |
|---|---|
| **C₁₇** |  |
| **C₁₈** |  |
| **C₁₉** |  |
| **C₂₀** |  |
| **C₂₁** |  |
| **C₂₂** |  |

**Scheme 1.** (*Continued.*)

8.05 (ddd, $J = 3.0, 6.0, 9.0$ Hz, 2H, Ar-H), 7.97 (d, $J = 9.0$ Hz, 2H, Ar-H),), 7.34–7.28 (m, 2H, Ar-H); $^{13}$C NMR (75 MHz, CDCl₃) $\delta$ 156.7, 156.5, 150.1, 143.1, 140.8, 137.5, 128.0, 125.8, 125.1, 124.5, 124.7, 123.6, 122.1, 120.0, 119.2, other carbons are isochronous.

## 3.11. [Zn(4'-(4-methylphenyl)-2,2':6',2''-terpyridine)(SO₄)] (**C₁**)

Light-purple amorphous solid; Yield: 85%; m.p. above 300°C; UV $\lambda_{\max}$ (CH₃CN) = 338 nm; $\lambda_{\text{em}}$ (CH₃CN): 405, 704 nm; FTIR (cm$^{-1}$): 1617 (C=N), 1478 (C=C), 1249 (C-N), 1112 (C-H), 587 (Zn-N); MS (MALDI-ToF) of [M]$^{+\cdot}$ : 387.0713.

| | |
|---|---|
| **C₂₃** |  |
| **C₂₄** |  |
| **C₂₅** |  |
| **C₂₆** |  |
| **C₂₇** |  |

**Scheme 1.** (*Continued.*)

## 3.12. [Co(4′-(4-methylphenyl)-2,2′:6′,2″-terpyridin)₂](PF₆)₂ (**C₂**)

Brick-red amorphous solid; Yield: 85%; m.p. above 300°C; UV $\lambda_{max}$ (CH₃CN) = 314 nm; $\lambda_{em}$ (CH₃CN): 379, 776 nm; FTIR (cm⁻¹): 1616 (C=N), 1471 (C=C), 1245 (C-N), 1110 (C-H), 623 (Co-N); MS (MALDI-ToF) of [M]⁺˙ : 705.2176.

## 3.13. [Fe(4′-(4-methylphenyl)-2,2′:6′,2″-terpyridin)₂] (PF₆)₂ (**C₃**)

Dark-purple amorphous solid; Yield: 79%; m.p. above 300°C; UV $\lambda_{max}$ (CH₃CN) = 327 nm; $\lambda_{em}$ (CH₃CN): 394, 742 nm; FTIR (cm⁻¹): 1611 (C=N), 1483 (C=C), 1286 (C-N), 1058 (C-H), 627 (Fe-N); MS (MALDI-ToF) of [M]⁺˙ : 702.2194.

## 3.14. [Zn(N,N-dimethyl-4-(2,2′:6′,2″-terpyridin-4′-yl)aniline)(SO₄)] (**C₄**)

Brown amorphous solid; Yield: 72%; m.p. above 300°C; UV $\lambda_{max}$ (CH₃CN) = 243 nm; $\lambda_{em}$ (CH₃CN): 370, 521, 732 nm; FTIR (cm⁻¹): 1595 (C=N), 1476 (C=C), 1250 (C-N), 1107 (C-H), 584 (Zn-N); MS (MALDI-ToF) of [M]⁺˙ : 416.0979.

### 3.15. [Co(4′-(N,N-dimethyl-4-(2,2′:6′,2″-terpyridin-4′-yl)aniline)$_2$](PF$_6$)$_2$ (C$_5$)

Candy-orange amorphous solid; Yield: 76%; m.p. above 300°C; UV $\lambda_{max}$ (CH$_3$CN) = 290 nm; $\lambda_{em}$ (CH$_3$CN): 547, 683 nm; FTIR (cm$^{-1}$): 1595 (C=N), 1473 (C=C), 1248 (C-N), 1125 (C-H), 625 (Co-N); MS (MALDI-ToF) of [M]$^{+\cdot}$: 763.2707.

### 3.16. [Fe(4′-(N,N-dimethyl-4-(2,2′:6′,2″-terpyridin-4′-yl)aniline)$_2$](PF$_6$)$_2$ (C$_6$)

Dark-purple amorphous solid; Yield: 77%; m.p. above 300°C; UV $\lambda_{max}$ (CH$_3$CN) = 324 nm; $\lambda_{em}$ (CH$_3$CN): 365, 638, 751 nm; FTIR (cm$^{-1}$): 1587 (C=N), 1468 (C=C), 1247 (C-N), 1090 (C-H), 629 (Fe-N); MS (MALDI-ToF) of [M]$^{+\cdot}$: 760.2725.

### 3.17. [Zn(N,N-diphenyl-4-(2,2′:6′,2″-terpyridin-4′-yl)aniline)(SO$_4$)] (C$_7$)

Brown amorphous solid; Yield: 80%; m.p. above 300°C; UV $\lambda_{max}$ (CH$_3$CN) = 428 nm; $\lambda_{em}$ (CH$_3$CN): 294, 359, 451, 717 nm; FTIR (cm$^{-1}$): 1660 (C=N), 1487 (C=C), 1286 (C-N), 1105 (C-H), 589 (Zn-N); MS (MALDI-ToF) of [M]$^{+\cdot}$: 540.1292.

### 3.18. [Co(N,N-diphenyl-4-(2,2′:6′,2″-terpyridin-4′-yl)aniline)$_2$](PF$_6$)$_2$ (C$_8$)

Marmalade (dark-yellow) amorphous solid; Yield: 70%; m.p. above 300°C; UV $\lambda_{max}$ (CH$_3$CN) = 308 nm; $\lambda_{em}$ (CH$_3$CN): 273, 453, 722 nm; FTIR (cm$^{-1}$): 1584 (C=N), 1506 (C=C), 1284 (C-N), 1050 (C-H), 621 (Co-N); MS (MALDI-ToF) of [M]$^{+\cdot}$: 1011.3310.

### 3.19. [Fe(N,N-diphenyl-4-(2,2′:6′,2″-terpyridin-4′-yl)aniline)$_2$](PF$_6$)$_2$ (C$_9$)

Maroon amorphous solid; Yield: 66%; m.p. above 300°C; UV $\lambda_{max}$ (CH$_3$CN) = 437 nm; $\lambda_{em}$ (CH$_3$CN): 341, 747 nm; FTIR (cm$^{-1}$): 1582 (C=N), 1505 (C=C), 1285 (C-N), 1150 (C-H), 627 (Fe-N); MS (MALDI-ToF) of [M]$^{+\cdot}$: 1008.3351.

### 3.20. [Zn(4′-(4-methoxyphenyl)-2,2′:6′,2″-terpyridine)(SO$_4$)] (C$_{10}$)

Greyish-purple amorphous solid; Yield: 78%; m.p. above 300°C; UV $\lambda_{max}$ (CH$_3$CN) = 286 nm; $\lambda_{em}$ (CH$_3$CN): 454, 856 nm; FTIR (cm$^{-1}$): 1600 (C=N), 1477 (C=C), 1295 (C-N), 1110 (C-H), 581 (Zn-N); MS (MALDI-ToF) of [M]$^{+\cdot}$: 403.0663.

### 3.21. [Co(4′-(4-methoxyphenyl)-2,2′:6′,2″-terpyridine)$_2$](PF$_6$)$_2$ (C$_{11}$)

Clay-brown amorphous solid; Yield: 66%; m.p. above 300°C; UV $\lambda_{max}$ (CH$_3$CN) = 286 nm; $\lambda_{em}$ (CH$_3$CN): 426, 704, 856 nm; FTIR (cm$^{-1}$): 1610 (C=N), 1473 (C=C), 1301 (C-N), 1098 (C-H), 625 (Co-N); MS (MALDI-ToF) of [M]$^{+\cdot}$: 737.2075.

### 3.22. [Fe(4′-(4-methoxyphenyl)-2,2′:6′,2″-terpyridine)$_2$](PF$_6$)$_2$ (C$_{12}$)

Dark-purple amorphous solid; Mol. Wt.: 1024.55 g mol$^{-1}$; Yield: 58%; m.p. above 300°C; UV $\lambda_{max}$ (CH$_3$CN) = 336 nm; $\lambda_{em}$ (CH$_3$CN): 409, 503, 759, 854 nm; FTIR (cm$^{-1}$): 3345 (C=N), 1607 (C=C), 1432 (C-N), 1057 (C-H), 628 (Fe-N); MS (MALDI-ToF) of [M]$^{+\cdot}$: 734.2092.

### 3.23. [Zn(4′-[3-(trifluoromethyl)phenyl]-2,2′:6′,2″-terpyridine)(SO$_4$)] (C$_{13}$)

Red-violet amorphous solid; Yield: 73%; m.p. above 300°C; UV $\lambda_{max}$ (CH$_3$CN) = 287 nm; $\lambda_{em}$ (CH$_3$CN): 853 nm; FTIR (cm$^{-1}$): 1616 (C=N), 1443 (C=C), 1326 (C-N), 1298 (C-H), 583 (Zn-N); MS (MALDI-ToF) of [M]$^{+\cdot}$: 441.0431.

### 3.24. [Co(4′-[3-(trifluoromethyl)phenyl]-2,2′:6′,2″-terpyridine)$_2$](PF$_6$)$_2$ (**C$_{14}$**)

Brick-red amorphous solid; Yield: 55%; m.p. above 300°C; UV $\lambda_{max}$ (CH$_3$CN) = 285 nm; $\lambda_{em}$ (CH$_3$CN): 405, 737, 850 nm; FTIR (cm$^{-1}$): 1617 (C=N), 1473 (C=C), 1327 (C-N), 1250 (C-H) 627 (Co-N); MS (MALDI-ToF) of [M]$^{+\cdot}$ : 813.1611.

### 3.25. [Fe(4′-[3-(trifluoromethyl)phenyl]-2,2′:6′,2″-terpyridine)$_2$](PF$_6$)$_2$ (**C$_{15}$**)

Dark-purple amorphous solid; Yield: 70%; m.p. above 300°C; UV $\lambda_{max}$ (CH$_3$CN) = 286 nm; $\lambda_{em}$ (CH$_3$CN): 403, 685, 850 nm; FTIR (cm$^{-1}$): 1612 (C=N), 1448 (C=C), 1327 (C-N), 1128 (C-H), 630 (Fe-N); MS (MALDI-ToF) of [M]$^{+\cdot}$ : 810.1629.

### 3.26. [Zn(4′-[2-(trifluoromethyl)phenyl]-2,2′:6′,2″-terpyridine)(SO$_4$)] (**C$_{16}$**)

Lilac (light purple) amorphous solid; Yield: 65%; m.p. above 300°C; UV $\lambda_{max}$ (CH$_3$CN) = 283 nm; $\lambda_{em}$ (CH$_3$CN): 401, 685, 851 nm; FTIR (cm$^{-1}$): 1617 (C=N), 1447 (C=C), 1327 (C-N), 1284 (C-H), 590 (Zn-N); MS (MALDI-ToF) of [M]$^{+\cdot}$ : 441.0431.

### 3.27. [Co(4′-[2-(trifluoromethyl)phenyl]-2,2′:6′,2″-terpyridine)$_2$](PF$_6$)$_2$ (**C$_{17}$**)

Brick-red amorphous solid; Yield: 78%; m.p. above 300°C; UV $\lambda_{max}$ (CH$_3$CN) = 285 nm; $\lambda_{em}$ (CH$_3$CN): 361, 735, 859 nm; FTIR (cm$^{-1}$): 1615 (C=N), 1476 (C=C), 1328 (C-N), 1197 (C-H), 627 (Co-N); MS (MALDI-ToF) of [M]$^{+\cdot}$ : 813.1611.

### 3.28. [Fe(4′-[2-(trifluoromethyl)phenyl]-2,2′:6′,2″-terpyridine)$_2$](PF$_6$)$_2$ (**C$_{18}$**)

Dark-purple amorphous solid; Yield: 71%; m.p. above 300°C; UV $\lambda_{max}$ (CH$_3$CN) = 286 nm; $\lambda_{em}$ (CH$_3$CN): 361, 580, 737, 865 nm; FTIR (cm$^{-1}$): 1612 (C=N), 1450 (C=C), 1328 (C-N), 1265 (C-H), 629 (Fe-N); MS (MALDI-ToF) of [M]$^{+\cdot}$ : 810.1629.

### 3.29. [Zn(4′-[4-(trifluoromethyl)phenyl]-2,2′:6′,2″-terpyridine)(SO$_4$)] (**C$_{19}$**)

Light-purple amorphous solid; Yield: 70%; m.p. above 300°C; UV $\lambda_{max}$ (CH$_3$CN) = 286 nm; $\lambda_{em}$ (CH$_3$CN): 409, 856 nm; FTIR (cm$^{-1}$): 1619 (C=N), 1448 (C=C), 1329 (C-N), 1198 (C-H), 592 (Zn-N); MS (MALDI-ToF) of [M]$^{+\cdot}$ : 441.0431.

### 3.30. [Co(4′-[4-(trifluoromethyl)phenyl]-2,2′:6′,2″-terpyridine)$_2$](PF$_6$)$_2$ (**C$_{20}$**)

Chocolate-brown amorphous solid; Yield: 65%; m.p. above 300°C; UV $\lambda_{max}$ (CH$_3$CN) = 289 nm; $\lambda_{em}$ (CH$_3$CN): 360, 730, 867 nm; FTIR (cm$^{-1}$): 1617 (C=N), 1480 (C=C), 1332 (C-N), 1267 (C-H), 623 (Co-N); MS (MALDI-ToF) of [M]$^{+\cdot}$ : 813.1611.

### 3.31. [Fe(4′-[4-(trifluoromethyl)phenyl]-2,2′:6′,2″-terpyridine)$_2$](PF$_6$)$_2$ (**C$_{21}$**)

Dark-purple amorphous solid; Yield: 66%; m.p. above 300°C; UV $\lambda_{max}$ (CH$_3$CN) = 286 nm; $\lambda_{em}$ (CH$_3$CN): 358, 578, 732, 869 nm; FTIR (cm$^{-1}$): 1615 (C=N), 1457 (C=C), 1334 (C-N), 1210 (C-H), 626 (Fe-N); MS (MALDI-ToF) of [M]$^{+\cdot}$ : 810.1629.

### 3.32. [Zn(4′-(3-nitrophenyl)-2,2′:6′,2″-terpyridine)(SO$_4$)] (**C$_{22}$**)

Light-purple amorphous solid; Yield: 74%; m.p. above 300°C; UV $\lambda_{max}$ (CH$_3$CN) = 347 nm; $\lambda_{em}$ (CH$_3$CN): 365, 680, 743, 855; FTIR (cm$^{-1}$): 1603 (C=N), 1467 (C=C), 1348 (C-N), 1073 (C-H), 585 (Zn-N); MS (MALDI-ToF) of [M]$^{+\cdot}$ : 310.040.

### 3.33. [Co(4′-(3-nitrophenyl)-2,2′:6′,2″-terpyridine)$_2$](PF$_6$)$_2$ (**C$_{23}$**)

Chocolate-brown amorphous solid; Yield: 65%; m.p. above 300°C; UV $\lambda_{max}$ (CH$_3$CN) = 334 nm; $\lambda_{em}$ (CH$_3$CN): 391, 699 nm; FTIR (cm$^{-1}$): 1614 (C = N), 1530 (C=C), 1246 (C-N), 1151 (C-H), 627 (Co-N); MS (MALDI-ToF) of [M]$^{+\cdot}$ : 767.1565.

### 3.34. [Fe(4′-(3-nitrophenyl)-2,2′:6′,2″-terpyridine)$_2$](PF$_6$)$_2$ (**C$_{24}$**)

Dark-purple amorphous solid; Yield: 70%; m.p. above 300°C; UV $\lambda_{max}$ (CH$_3$CN) = 328 nm; $\lambda_{em}$ (CH$_3$CN): 402, 684, 856 nm; FTIR (cm$^{-1}$): 1614 (C=N), 1474 (C=C), 1351 (C-N), 1025 (C-H), 629 (Fe-N); MS (MALDI-ToF) of [M]$^{+\cdot}$ : 764.1584.

### 3.35. [Zn(4′-(4-nitrophenyl)-2,2′:6′,2″-terpyridine)(SO$_4$)] (**C$_{25}$**)

Caramel (brown) amorphous solid; Yield: 77%; m.p. above 300°C; UV $\lambda_{max}$ (CH$_3$CN) = 287 nm; $\lambda_{em}$ (CH$_3$CN): 474, 850 nm; FTIR (cm$^{-1}$): 1619 (C=N), 1476 (C=C), 1359 (C-N), 1167 (C-H), 587 (Zn-N); MS (MALDI-ToF) of [M]$^{+\cdot}$ : 310.040.

### 3.36. [Co(4′-(4-nitrophenyl)-2,2′:6′,2″-terpyridine)$_2$](PF$_6$)$_2$ (**C$_{26}$**)

Chocolate-brown amorphous solid; Yield: 76%; m.p. above 300°C; UV $\lambda_{max}$ (CH$_3$CN) = 289 nm; $\lambda_{em}$ (CH$_3$CN): 428, 563, 850 nm; FTIR (cm$^{-1}$): 1616 (C=N), 1471 (C=C), 1245 (C-N), 1245 (C-H), 628 (Co-N); MS (MALDI-ToF) of [M]$^{+\cdot}$ : 767.1565.

### 3.37. [Fe(4′-(4-nitrophenyl)-2,2′:6′,2″-terpyridine)$_2$](PF$_6$)$_2$ (**C$_{27}$**)

Dark-purple amorphous solid; Yield: 83%; m.p. above 300°C; UV $\lambda_{max}$ (CH$_3$CN) = 287 nm; $\lambda_{em}$ (CH$_3$CN): 565, 848 nm; FTIR (cm$^{-1}$): 1618 (C=N), 1472 (C=C), 1350 (C-N), 1287 (C-H), 630 (Fe-N); MS (MALDI-ToF) of [M]$^{+\cdot}$ : 764.1598.

## 3.38. Absorption properties

With a broad class of terpyridine compounds in hand, we examined their initial photophysical properties. The UV-Vis spectra of terpyridines and their metal complexes are recorded in chloroform and acetonitrile solutions, respectively, and the related data are given in table 1. In the absorption spectra recorded in chloroform for the ligands (**L$_1$**–**L$_9$**), strong, broad absorption bands appear in the range 234–325 nm, while for the complexes (**C$_1$**–**C$_{27}$**), the absorption bands were seen in the range 240–429 nm. All these absorptions are attributed to π–π* electronic transitions in the aromatic system of the terpyridines. The absorption maxima for **C$_1$**–**C$_{27}$** are somewhat shifted from those of the free ligands by affixing either electron-releasing or -withdrawing groups at the central pyridine. The longest absorption wavelength in the **L$_1$** UV-Vis spectrum ($\lambda_{max}$ = 325 nm) is broad and comparable to that of the un-substituted parent terpyridine (table 1). The UV-Vis spectra of the Zn(II), Co(II) and Fe(II) complexes in acetonitrile solutions are shown in electronic supplementary material, table S1. The variation in absorption wavelengths and intensities is the result of different substituents in the 4′ position of the terpyridine. The UV-Vis spectra of terpyridine-based metal complexes show the characteristic metal–ligand charge transfer (MLCT) shift of the Fe(II)-terpyridine system (**C$_3$**, **C$_6$**, **C$_9$**, **C$_{12}$**, **C$_{15}$**, **C$_{18}$**, **C$_{21}$**, **C$_{24}$**, **C$_{27}$**) at $\lambda_{max}$ = 587 nm and the bands at 284 nm and 426 nm are due to ligand-centred (LC) transitions. A slight redshift when related to the reference iron (II) terpyridine complex has been detected. The spin allowed the MLCT band in the visible region to undergo a rise in intensity and a redshift, regardless of electron-donor or -acceptor nature of the substituents. The intense absorption bands of the Zn(II) complexes (**C$_1$**, **C$_4$**, **C$_7$**, **C$_{10}$**, **C$_{13}$**, **C$_{16}$**, **C$_{19}$**, **C$_{22}$**, **C$_{25}$**) were seen in the range 215–586 nm. These absorption peaks are mainly due to the metal–ligand charge transfer of π–π* and n–π* transitions having $\lambda_{max}$ = 586 nm. This redshift could be credited to a decrease in the LUMO energy values of the complexes. The absorption bands of the Co(II) complexes (**C$_2$**, **C$_5$**, **C$_8$**, **C$_{11}$**, **C$_{14}$**, **C$_{17}$**, **C$_{20}$**, **C$_{23}$**, **C$_{26}$**) were seen in the range 209–571 nm and their MLCT transitions have a maximum value at 530 nm.

Surprisingly, the electron-donating group *p*-methoxyphenyl at the 4′-position of the terpyridine ring in Zn(II) and Co(II) complexes shifts $\lambda_{max}$ towards longer wavelengths, resulting in a 30 nm bathochromic

**Table 1.** Absorption and emission data of all the synthesized compounds.

| compound code | solvent | excitation wavelength (nm) | $\lambda_{em}$ (nm) | $\lambda_{max}$ (nm) | transitions | MLCT band | Stokes shift (nm) |
|---|---|---|---|---|---|---|---|
| $L_1$ | chloroform | 335 | 357, 731 | 325 | $\pi-\pi^*$ (LC) | — | 374 |
| $L_2$ | chloroform | 303 | 344, 443 | 293 | $\pi-\pi^*$ (LC) | — | 109 |
| $L_3$ | chloroform | 243 | 335, 576 | 234 | $\pi-\pi^*$ (LC) | — | 241 |
| $L_4$ | chloroform | 258 | 371, 746, 855 | 248 | $\pi-\pi^*$ (LC) | — | 109 |
| $L_5$ | chloroform | 243 | 359, 727, 861 | 234 | $\pi-\pi^*$ (LC) | — | 134 |
| $L_6$ | chloroform | 250 | 345, 70, 85 | 240 | $\pi-\pi^*$ (LC) | — | 15 |
| $L_7$ | chloroform | 243 | 345, 723, 852 | 234 | $\pi-\pi^*$ (LC) | — | 129 |
| $L_8$ | chloroform | 263 | 361, 723, 849 | 253 | $\pi-\pi^*$ (LC) | — | 126 |
| $L_9$ | chloroform | 257 | 403, 720, 851 | 247 | $\pi-\pi^*$ (LC) | — | 131 |
| $C_1$ | acetonitrile | 348 | 405, 704 | 338 | $\pi-\pi^*$ (C=C), $n-\pi^*$ (C=N) | 575 (d$\pi \rightarrow \pi^*$) | 299 |
| $C_2$ | acetonitrile | 324 | 379, 776 | 314 | $\pi-\pi^*$ (C=C), $n-\pi^*$ (C=N) | 523 (d$\pi \rightarrow \pi^*$) | 397 |
| $C_3$ | acetonitrile | 337 | 394, 742 | 327 | $\pi-\pi^*$ (C=C), $n-\pi^*$ (C=N) | 571 (d$\pi \rightarrow \pi^*$) | 348 |
| $C_4$ | acetonitrile | 253 | 370, 521, 732 | 243 | $\pi-\pi^*$ (C=C), $n-\pi^*$ (C=N) | 580 (d$\pi \rightarrow \pi^*$) | 211 |
| $C_5$ | acetonitrile | 300 | 547, 683 | 290 | $\pi-\pi^*$ (C=C), $n-\pi^*$ (C=N) | 442 (d$\pi \rightarrow \pi^*$) | 136 |
| $C_6$ | acetonitrile | 334 | 365, 638, 751 | 324 | $\pi-\pi^*$ (C=C), $n-\pi^*$ (C=N) | 587 (d$\pi \rightarrow \pi^*$) | 113 |
| $C_7$ | acetonitrile | 438 | 294, 359, 451, 717 | 428 | $\pi-\pi^*$ (C=C), $n-\pi^*$ (C=N) | 583 (d$\pi \rightarrow \pi^*$) | 266 |
| $C_8$ | acetonitrile | 318 | 273, 453, 722 | 308 | $\pi-\pi^*$ (C=C), $n-\pi^*$ (C=N) | 447 (d$\pi \rightarrow \pi^*$) | 269 |
| $C_9$ | acetonitrile | 447 | 341, 747 | 437 | $\pi-\pi^*$ (C=C), $n-\pi^*$ (C=N) | 584 (d$\pi \rightarrow \pi^*$) | 406 |
| $C_{10}$ | acetonitrile | 296 | 454, 856 | 286 | $\pi-\pi^*$ (C=C), $n-\pi^*$ (C=N) | 586 (d$\pi \rightarrow \pi^*$) | 402 |
| $C_{11}$ | acetonitrile | 296 | 426, 704, 856 | 286 | $\pi-\pi^*$ (C=C), $n-\pi^*$ (C=N) | 538 (d$\pi \rightarrow \pi^*$) | 152 |
| $C_{12}$ | acetonitrile | 346 | 409, 503, 759, 854 | 336 | $\pi-\pi^*$ (C=C), $n-\pi^*$ (C=N) | 586 (d$\pi \rightarrow \pi^*$) | 95 |
| $C_{13}$ | acetonitrile | 297 | 853 | 287 | $\pi-\pi^*$ (C=C), $n-\pi^*$ (C=N) | 570 (d$\pi \rightarrow \pi^*$) | 568 |
| $C_{14}$ | acetonitrile | 295 | 405, 737, 850 | 285 | $\pi-\pi^*$ (C=C), $n-\pi^*$ (C=N) | 526 (d$\pi \rightarrow \pi^*$) | 113 |

**Table 1.** (*Continued.*)

| compound code | solvent | excitation wavelength (nm) | $\lambda_{em}$ (nm) | $\lambda_{max}$ (nm) | transitions | MLCT band | Stokes shift (nm) |
|---|---|---|---|---|---|---|---|
| **C$_{15}$** | acetonitrile | 296 | 403, 685, 850 | 286 | $\pi$–$\pi^*$ (C=C), n–$\pi^*$ (C=N) | 568 (d$\pi$ → $\pi^*$) | 165 |
| **C$_{16}$** | acetonitrile | 293 | 401, 685, 851 | 283 | $\pi$–$\pi^*$ (C=C), n–$\pi^*$ (C=N) | 569 (d$\pi$ → $\pi^*$) | 166 |
| **C$_{17}$** | acetonitrile | 295 | 361, 735, 859 | 285 | $\pi$–$\pi^*$ (C=C), n–$\pi^*$ (C=N) | 538 (d$\pi$ → $\pi^*$) | 124 |
| **C$_{18}$** | acetonitrile | 296 | 361, 580, 737, 865 | 286 | $\pi$–$\pi^*$ (C=C), n–$\pi^*$ (C=N) | 572 (d$\pi$ → $\pi^*$) | 128 |
| **C$_{19}$** | acetonitrile | 296 | 409, 856 | 286 | $\pi$–$\pi^*$ (C=C), n–$\pi^*$ (C=N) | 540 (d$\pi$ → $\pi^*$) | 447 |
| **C$_{20}$** | acetonitrile | 299 | 360, 730, 867 | 289 | $\pi$–$\pi^*$ (C=C), n–$\pi^*$ (C=N) | 435 (d$\pi$ → $\pi^*$) | 137 |
| **C$_{21}$** | acetonitrile | 296 | 358, 578, 733, 869 | 286 | $\pi$–$\pi^*$ (C=C), n–$\pi^*$ (C=N) | 569 (d$\pi$ → $\pi^*$) | 138 |
| **C$_{22}$** | acetonitrile | 357 | 365, 680, 743, 855 | 347 | $\pi$–$\pi^*$ (C=C), n–$\pi^*$ (C=N) | 580 (d$\pi$ → $\pi^*$) | 112 |
| **C$_{23}$** | acetonitrile | 344 | 391, 699 | 334 | $\pi$–$\pi^*$ (C=C), n–$\pi^*$ (C=N) | 535 (d$\pi$ → $\pi^*$) | 308 |
| **C$_{24}$** | acetonitrile | 338 | 402, 684, 856 | 328 | $\pi$–$\pi^*$ (C=C), n–$\pi^*$ (C=N) | 575 (d$\pi$ → $\pi^*$) | 172 |
| **C$_{25}$** | acetonitrile | 297 | 474, 850 | 287 | $\pi$–$\pi^*$ (C=C), n–$\pi^*$ (C=N) | 581 (d$\pi$ → $\pi^*$) | 376 |
| **C$_{26}$** | acetonitrile | 299 | 428, 563, 850 | 289 | $\pi$–$\pi^*$ (C=C), n–$\pi^*$ (C=N) | 545 (d$\pi$ → $\pi^*$) | 287 |
| **C$_{27}$** | acetonitrile | 297 | 565, 848 | 287 | $\pi$–$\pi^*$ (C=C), n–$\pi^*$ (C=N) | 583 (d$\pi$→$\pi^*$) | 283 |

LC = ligand centred transition.

MLCT = metal to ligand charge transfer transition.

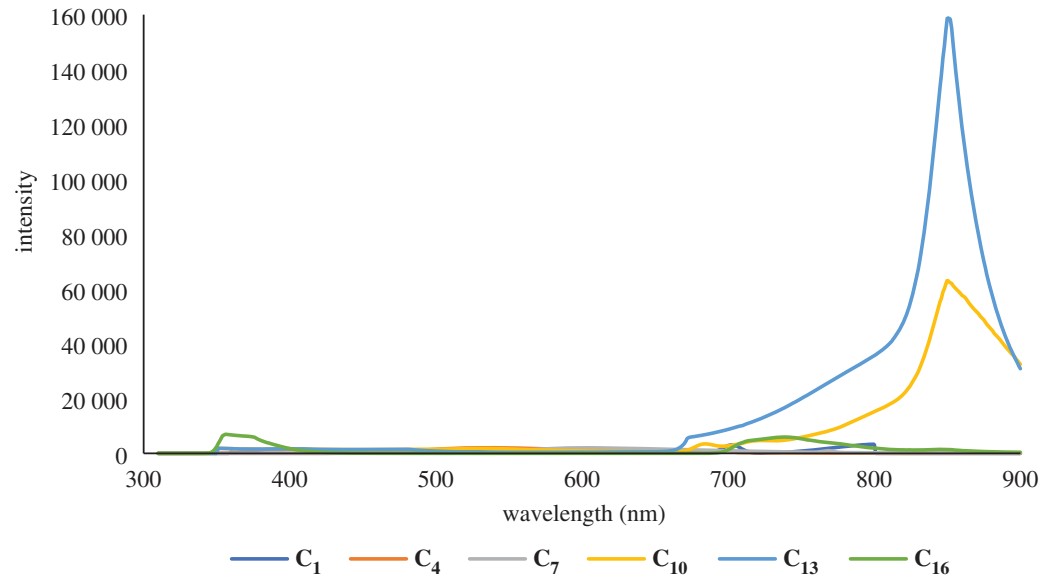

**Figure 2.** A comparison of emission spectra of Zn complexes of **L₁–L₆** in solution form.

shift. This result supports the concept that the MLCT state can be higher in energy by nitration of the terpyridine ligand-based metal complexes, the 4-methoxyphenyl group at the 4′-position of the terpyridine ring in the complexes have a shorter wavelength, and a significant maximum enhancement in MLCT band energy was observed. The absorption bands are influenced significantly by the aryl substituents on the terminal phenyl moiety. Generally, the absorption spectra of all complexes are typically well correlated with those found for the respective uncomplexed ligands.

## 3.39. Photo-luminescent properties

Attaching various functional groups to terpyridine at the *p*-position may not only open several paths towards the development of advanced supramolecular structures but may also influence the fluorescence properties of the compounds. Due to their potential applications in chemical sensors, electroluminescent displays and fluorescent materials, inorganic-organic hybrid complexes, particularly those with $d^{10}$ transition metal centres, are now of great interest. From the structural analysis, the ligand can be seen to have a large *p*-conjugated aromatic system that can possess strong fluorescence and efficient transfer of energy. The photo-luminescence spectra of the Zn, Co and Fe (**C₁–C₂₇**) complexes were recorded in acetonitrile solutions at ambient temperature with an excitation *λ* of 320 nm for the Zn(II) complexes, 286 nm for the Co complexes and 295 nm for the Fe complexes. All **C₁–C₂₇** complexes display emission bands in the visible region of the electromagnetic spectrum. The maximum emission peaks for the Zn, Co and Fe complexes were observed at 857, 859 and 865 nm, respectively, as described in table 1. The compounds having the lowest energy-absorption bands, which are below 310 nm, are not much different from the parent compounds. Fluorescence of **L₁–L₉** was observed at about 340 nm irrespective of the location of the substitution and the number of phenyl groups. The excitation spectrum observed between 330 and 400 nm at any stage was similar to the corresponding absorption spectra, indicating that the excitation of the lowest energy absorption of **L₁–L₉** led to the emitting state. In the *p*-position of the 4-phenyl complex, several substituents with different electron-donating or -withdrawing properties were added to further analyse and modify the fluorescence properties of 4-phenyl terpyridine. The *p*-substituted terpyridine complexes **C₁–C₂₇** absorption and fluorescence maxima are listed in table 1.

The photo-luminescent properties of compounds **C₁–C₄** in solution and at ambient temperature are characterized by the fluorescence spectra presented in figures 2–4 and supporting information file. Compound **C₁** has two bands in its emission spectrum, a lower intensity one at *ca* 405 nm and another with higher intensity at *ca* 704 nm. Similarly, compound **C₂** has two bands in its emission spectra, a low-intensity one at *ca* 379 nm and another with higher intensity at *ca* 776 nm. Compared with the previously reported [66] photo-luminescence of the ligand **L₁**, the higher energy bands of **L₁** and **L₂** may tentatively be assigned to an intra-ligand charge transfer (ILCT), while the latter band of **C₁** is assumed to be the ligand–metal charge transfer (LMCT) and that of **C₂** to a ligand-based intramolecular

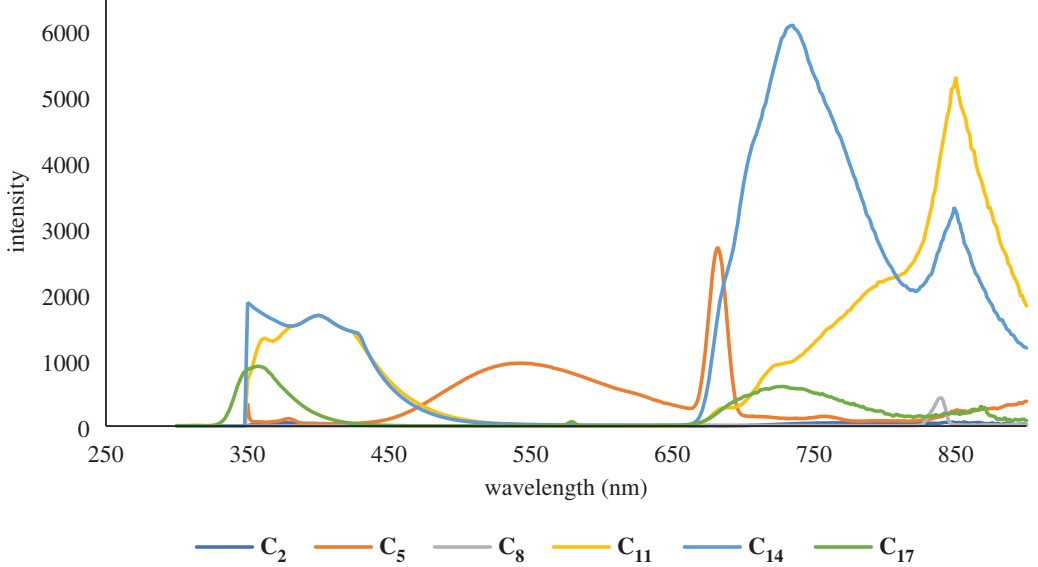

**Figure 3.** A comparison of emission spectra of Co complexes of $L_1$–$L_6$ in solution form.

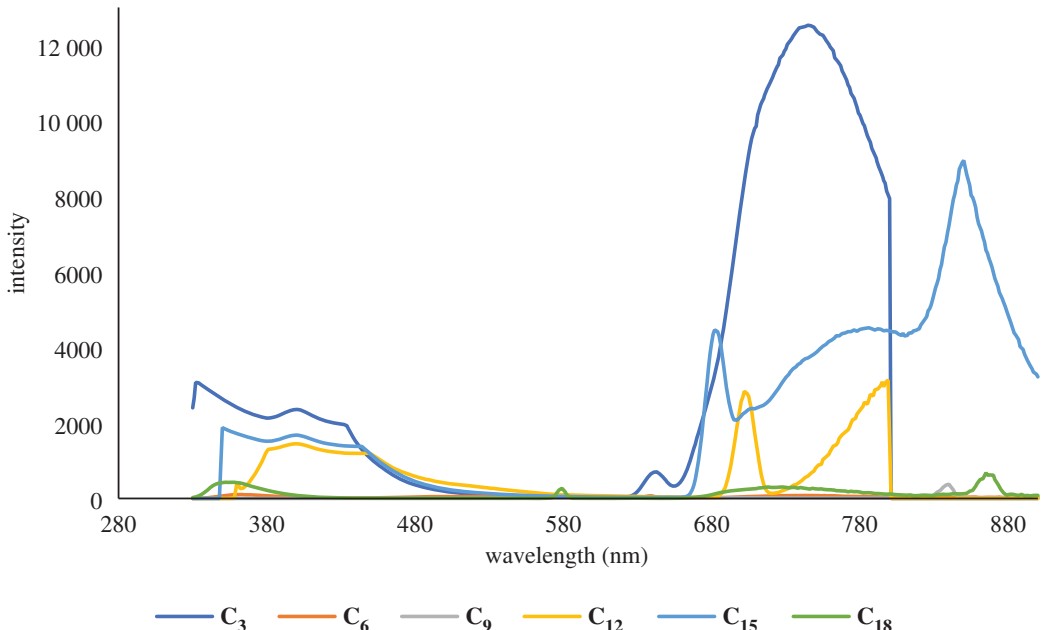

**Figure 4.** A comparison of emission spectra of Fe complexes of $L_1$–$L_6$ in solution state.

charge transfer (ICT). In $C_3$, the narrow band is present at 394 nm and broad band at 742 nm is also seen. This band is linked to an ILCT based on its energy that is similar to that recorded for a related $FeSO_4$ complex, which shows three bands, one at 402 nm, and the other two bands at 684 and 856 nm, respectively. The higher energy ILCT band (at 402 nm in $C_3$) is not sensitive to the transition from a 3d to a 4d metal complex (in the same group). The $C_4$ emission spectrum also displays three bands, one strong at 370 nm, one of low intensity at approximately 521 nm and a broad band at 732 nm. According to the results noted above, the former peak is tentatively assigned to an ILCT transition and the latter to an LMCT transition. From the structural analysis, it can be seen that ligand has a large π-conjugated aromatic system, which may possess strong fluorescence and efficient energy transfer. The free ligand L exhibits intense fluorescence with emission maxima at 372 and 522 nm upon excitation at 300 nm, while complexes show bright fluorescence with maximum emission at 547 and 683 nm for $C_5$, 365 nm for $C_6$ and 359 nm for $C_7$ upon excitation at 312 nm. The emission values for complexes $C_8$–$C_{27}$ were in the range of 453 to 856 nm upon excitation at about 340 nm. The difference in emissions between the

ligands and the complexes is due to the various energy transfer routes, and the emissions detected in the complexes may be ascribed to the intra-ligand $\pi$–$\pi^*$ transitions.

The different substituents on terpyridine moiety in the Co(II) complexes generate a wider range for their emission spectra (365–850 nm) than for the Zn(II) and Fe(II) complexes. Substitution of relatively low electron-donating or withdrawal groups, such as methyl, only had a slight effect on the overall fluorescence properties.

## 3.40. Anti-microbial evaluation

Many microorganisms are becoming immune to traditional antibiotics and thus there is a need to introduce new antibiotics against those pathogens. In this regard, all the new compounds were assessed for their *in vitro* anti-bacterial activity against Gram-positive bacteria *Staphylococcus aureus* (ATCC 6538) and two Gram-negative bacteria *Escherichia coli* (ATCC 25922) and *Pseudomonas aeruginosa* (ATCC 9721). In addition, these compounds were evaluated for their anti-fungal activity against *Candida albicans* (ATCC 9002) and *Candida parapsilosis* (ATCC 22019) fungal strains. The results were recorded as the average diameter of ZOI of fungal or bacterial growth around the plates in millimetres for each compound tested. Both anti-bacterial and anti-fungal results are displayed in table 2.

## 3.41. Anti-bacterial activity

Anti-bacterial activity of the compounds has been checked against three bacterial strains including *P. aeruginosa, S. aureus, E. coli* using a well diffusion method. Henceforth, the relative screening of anti-bacterial activity of the new terpyridine-based ligands, $L_1$ to $L_9$, and their metal complexes, $C_1$ to $C_{27}$, have been carried out on Gram-positive and Gram-negative bacteria. Cefixime was used as a standard. An assessment of anti-bacterial activity among all the compounds showed them to be very potent against *P. aeruginosa* and *E. coli*. The free ligand has considerable activity against *P. aeruginosa* and *E. coli* (ZOI $\geq$ 20 mm) but has moderate activity against *S. aureus* (ZOI $\leq$ 17 mm). In contrast with the free terpyridine ligand, the complex has more potent activity against *E. coli* (more ZOI), but it is less active against *S. aureus*. *In vitro*, anti-bacterial studies showed that all the complexes showed varying degrees of inhibition. Among the metal complexes, $C_1$ displayed excellent activity against *E. coli* with ZOI = 23.7 mm while $C_8$, having ZOI = 20.7 mm, showed better activity compared with $C_{22}$ and $C_{23}$ having ZOI = 18.8 and 17.8 mm, respectively. In the case of *P. aeruginosa*, complexes $C_{24}$, $C_8$ and $C_6$ showed potential IZ diameters of 18.5, 17.3 and 17.2 mm respectively, as compared with complexes $C_3$ and $C_1$. However, the compounds under examination were found to be almost inactive against *Staphylococcus aureus*, except $C_1$, $C_{22}$ and $C_{24}$, which showed ZOIs of about 16.5, 14.5 and 15.7 mm, respectively. Thus, all-metal complexes showed significant ZOIs against *P. Aeruginosa* and *E. coli* as compared with *S. aureus*. A comparative review of anti-bacterial findings indicated that the metal complexes displayed improved anti-bacterial activity compared with free ligands, as previously reported, from $L_1$ to $L_9$. The remaining members of this series displayed moderate to good anti-bacterial activity and are comparatively more active against *P. aeruginosa* and *E. Coli* bacteria than *S. aureus* strain.

## 3.42. Anti-fungal activity

The new compounds were also evaluated against the two different fungal strains *C. albicans* and *C. parapsilosis*. The outcomes of these anti-fungal activities have been presented in table 2. The standard drug used was clotrimazole. Terpyridine analogues ($L_1$–$L_9$) were found less active as compared with the standard. Again, the compounds $C_{24}$, $C_{16}$ and $C_8$, having ZOI values of 21.6, 19.8 and 19.5 mm, respectively, exhibited the highest activity against both fungal strains. However, these compounds were found comparatively more potent against fungi as compared with the bacteria. Although with respect to the standard drug, all the evaluated compounds were found to show moderate to good activity against both *C. albicans* and *C. parapsilosis* fungi.

## 3.43. Structure–activity and structure–property relationships

As far as absorption behaviour is concerned, the absorption bands are influenced significantly by the aryl substituents on the terminal aryl moiety. Compared with the phenyl-substituted compound, electron-

**Table 2.** Anti-microbial assay (ZOI) of all the synthesized compounds.

| compound number | zones of inhibition (mm) | | | | |
| --- | --- | --- | --- | --- | --- |
| | anti-bacterial activity | | | anti-fungal activity | |
| | P. aeruginosa | S. aureus | E. coli | C. albicans | C. parapsilosis |
| L₁ | 0 | 0 | 14.5 ± 0.7 | 11.8 ± 0.7 | 0 |
| L₂ | 15.1 ± 0.7 | 0 | 16.5 ± 0.4 | 12 ± 0.1 | 0 |
| L₃ | 0 | 0 | 16.5 ± 0.1 | 0 | 0 |
| L₄ | 0 | 0 | 15.5 ± 0.2 | 19.3 ± 0.3 | 14.6 ± 0.1 |
| L₅ | 0 | 0 | 17.5 ± 0.3 | 17.3 ± 0.2 | 17.5 ± 0.7 |
| L₆ | 14.3 ± 0.3 | 0 | 0 | 17.4 ± 0.5 | 17.5 ± 0.4 |
| L₇ | 18.1 ± 0.4 | 0 | 0 | 15.3 ± 0.2 | 0 |
| L₈ | 0 | 0 | 0 | 0 | 14.7 ± 0.8 |
| L₉ | 0 | 0 | 18.7 ± 0.7 | 0 | 11.6 ± 0.6 |
| C₁ | 16.3 ± 0.7 | 16.5 ± 0.7 | 23.7 ± 0.5 | 17.3 ± 0.6 | 0 |
| C₂ | 16.5 ± 0.7 | 0 | 20.8 ± 0.5 | 18.9 ± 0.6 | 0 |
| C₃ | 16.7 ± 0.1 | 0 | 16.7 ± 0.6 | 11.4 ± 0.5 | 18.7 ± 0.3 |
| C₄ | 17.9 ± 0.3 | 18.5 ± 0.7 | 15.4 ± 0.3 | 0 | 0 |
| C₅ | 17.2 ± 0.3 | 13.5 ± 0.7 | 15.7 ± 0.3 | 0 | 14.7 ± 0.3 |
| C₆ | 17.2 ± 0.3 | 0 | 15.7 ± 0.3 | 0 | 0 |
| C₇ | 19.3 ± 0.1 | 12.5 ± 0.4 | 21.7 ± 0.4 | 0 | 14.6 ± 0.6 |
| C₈ | 17.3 ± 0.1 | 14.5 ± 0.4 | 20.7 ± 0.4 | 19.5 ± 0.7 | 15.6 ± 0.6 |
| C₉ | 0 | 0 | 0 | 0 | 12.6 ± 0.5 |
| C₁₀ | 17.3 ± 0.2 | 15.7 ± 0.5 | 14.8 ± 0.2 | 0 | 0 |
| C₁₁ | 14.2 ± 0.2 | 0 | 13.8 ± 0.3 | 18.8 ± 0.6 | 0 |
| C₁₂ | 11.5 ± 0.4 | 17.5 ± 0.4 | 0 | 12.6 ± 0.6 | 18.3 ± 0.7 |
| C₁₃ | 14.3 ± 0.3 | 15.7 ± 0.3 | 18.8 ± 0.5 | 0 | 15.8 ± 0.8 |
| C₁₄ | 18.2 ± 0.2 | 0 | 19.8 ± 0.3 | 13.8 ± 0.6 | 18.6 ± 0.9 |
| C₁₅ | 0 | 0 | 0 | 0 | 15.7 ± 0.7 |
| C₁₆ | 19.3 ± 0.3 | 16.8 ± 0.5 | 12.8 ± 0.5 | 19.8 ± 0.1 | 18.8 ± 0.6 |
| C₁₇ | 12.2 ± 0.2 | 0 | 0 | 15.8 ± 0.6 | 12.4 ± 0.5 |
| C₁₈ | 14.5 ± 0.4 | 0 | 0 | 0 | 15.3 ± 0.7 |
| C₁₉ | 18.3 ± 0.3 | 15.7 ± 0.3 | 18.8 ± 0.5 | 15.8 ± 0.1 | 0 |
| C₂₀ | 0 | 0 | 16.8 ± 0.3 | 0 | 14.6 ± 0.5 |
| C₂₁ | 18.5 ± 0.4 | 0 | 0 | 21.6 ± 0.6 | 0 |
| C₂₂ | 14.3 ± 0.3 | 15.7 ± 0.3 | 14.8 ± 0.5 | 15.8 ± 0.1 | 16.8 ± 0.6 |
| C₂₃ | 14.2 ± 0.2 | 0 | 19.8 ± 0.3 | 15.8 ± 0.6 | 10.6 ± 0.5 |
| C₂₄ | 18.5 ± 0.4 | 0 | 0 | 21.6 ± 0.6 | 15.3 ± 0.7 |
| C₂₅ | 19.3 ± 0.3 | 14.7 ± 0.3 | 17.8 ± 0.5 | 15.8 ± 0.1 | 12.8 ± 0.6 |
| C₂₆ | 16.2 ± 0.2 | 0 | 15.8 ± 0.3 | 18.8 ± 0.6 | 0 |
| C₂₇ | 17.5 ± 0.4 | 18.7 ± 0.3 | 0 | 0 | 11.3 ± 0.7 |
| negative (DMSO) | — | — | — | — | — |
| standard (cefixime) | 28.7 ± 0.3 | 27.5 ± 0.7 | 30.5 ± 0.3 | — | — |
| standard (clotrimazole) | — | — | — | 28.9 ± 0.8 | 25.6 ± 0.6 |

withdrawing substituents such as the 4-CF$_3$ group cause noticeable hypsochromic and electron-donating substituents like $-N(CH_3)_2$, $-N(Ph)_2$ cause noticeable bathochromic shifts of the $\pi$–$\pi^*$ band, respectively.

The photo-luminescent properties of compounds $C_1$–$C_{27}$ (in acetonitrile solution) have been investigated and figures 2–4 present the emission spectroscopic data. Likewise, table 1 presents the emission bands. Compounds carrying groups with varying electro-negativity exhibit varying photo-luminescent properties in solution at room temperature. When the substituent groups are modified, the average emission levels of these compounds can be estimated from 273 to 865 nm. Only a single band is noticed in the fluorescence (emission) spectrum of complex $C_{13}$, while two bands are detected in compounds $C_1$, $C_2$, $C_3$, $C_5$, $C_9$, $C_{11}$, $C_{19}$, $C_{23}$, $C_{25}$ and $C_{27}$, three bands are detected in compounds $C_4$, $C_6$, $C_8$, $C_{14}$, $C_{15}$, $C_{16}$, $C_{17}$, $C_{20}$ and $C_{26}$ and four bands are detected in compounds $C_7$, $C_{18}$, $C_{21}$ and $C_{22}$. In $C_{13}$, only a broad band at 853 nm is seen when excited at 295 nm. For compounds $C_1$, $C_2$, $C_3$, $C_5$, $C_9$, $C_{11}$, $C_{19}$, $C_{23}$, $C_{25}$ and $C_{27}$, their photo-luminescent spectra display two bands, a low-intensity band at 405 nm and another with higher intensity at 704 nm for $C_1$, 379 and 776 nm for $C_2$, 394 and 792 nm for $C_3$, 547 and 683 nm for $C_5$, 341 and 747 nm for $C_9$, 454 and 856 nm for $C_{10}$, 409 and 856 nm for $C_{19}$, 391 and 699 nm for $C_{23}$, 474 and 850 nm for $C_{25}$ and 565 and 848 nm for $C_{27}$,respectively, when being excited at around 320 nm. Compounds $C_4$, $C_6$, $C_8$, $C_{11}$, $C_{12}$, $C_{14}$, $C_{15}$, $C_{16}$, $C_{17}$, $C_{20}$, $C_{24}$ and $C_{26}$ show multiple emission behaviours in their spectra, with emission maxima at around 273, 630 and 859 nm upon excitation at 300 nm, whereas the complexes $C_7$, $C_{18}$, $C_{21}$ and $C_{22}$ show maximal fluorescence with four emission bands at 294, 359, 451 and 717 nm for $C_7$, 361, 580, 737 and 865 nm for $C_{18}$, 358, 578, 732 and 869 nm for $C_{21}$ and 365, 680, 743 and 855 nm for $C_{22}$ upon excitation at 305 nm. On account of the different substituents at the terpyridine unit, their emission peaks show variations. All of them not only exhibit photo-luminescent properties with high emission but also change the maximum emission peaks from 305 to 856 nm due to the substituent groups.

A strong absorption band for $C_1$–$C_{27}$ is due to the $\pi$–$\pi^*$ transitions of a conjugate backbone with maximum absorption $\lambda$ in the range of 300–869 nm. Moreover, due to the presence of a similar central terpyridine nucleus for all the compounds, the spectral shapes/curves are identical in the UV-Vis (absorption) spectra. It is noteworthy, that the difference in wavelengths and strength of absorption is attributable solely to the influence of the 4-alkylphenyl moieties at the 4'-position of 2,2':6'',2 terpyridine nucleus. The photo-luminescent properties of the complexes ($C_1$–$C_{27}$) were then analysed at their corresponding wavelengths of excitation. Compounds having maximum bands showed a maximum emission at higher wavelengths relative to other compounds in the series. This common trend can be attributed to the various configurations of the alkyl groups in solution. These complexes ($C_1$–$C_{27}$) also display blue fluorescence due to the presence of their maximum emission wavelengths in the blue region of the visible spectrum, close to most previously recorded terpyridine complexes [65]. The terpyridine ring's structure and substitution pattern are thought to be responsible for certain compounds' blue fluorescence. Aryl groups that exist at $p$-position terpyridine form the conjugated backbone and are primarily responsible for photon absorption. As a result, the substitution of alkyl-aryl groups at 4-position can be assumed to have some effect on physico-chemical properties. It is apparent from the results that changing the length of the alkyl group substantially affects emission strength without a large red- or blue-shift in the emission wavelength, and the replacement of the electron-donating groups (EDG) for the conjugated backbone of these compounds can be used to modify the emission intensity. It is therefore anticipated that by adjusting the 4-aryl terpyridine ring substituent, the emission intensity can be changed, which is very useful for controlling the luminescence and optoelectronic properties of the organic light-emitting diodes (OLEDs) based on related fluorescent compounds.

The SAR studies revealed that the different substituents on ligands are responsible for controlling bioactivities (figure 5). The finding regarding biological aspects of metal complexes shows that the electron-withdrawal substituent on the ligand will boost the anti-bacterial function of the original compound, while the case is reversed for electron-releasing substituents. Such a pattern indicates that the substituents can efficiently regulate the complex's anti-bacterial function. Bulky benzene ring substituents, such as a lipophilic methyl group, have not enhanced the biological activity of parent's analogues. This supports the favourable electron-withdrawal substituent at position 4 to allow the complex structure to become more active. The strong anti-microbial activity observed may be due to the metal centre's nuclearity in the complex; dinuclear centres are usually more active than mononuclear ones. Similar to metal salts and the corresponding ligands, both complexes have anti-bacterial activity against *P. aeruginosa* and *E. coli*, which can be linked to the concept of chelation, where the chelation of metal ions and ligands decreases the polarity of the metal ion mainly due to the partial sharing of its positive charge with the donor groups and potential $p$-electron delocalization

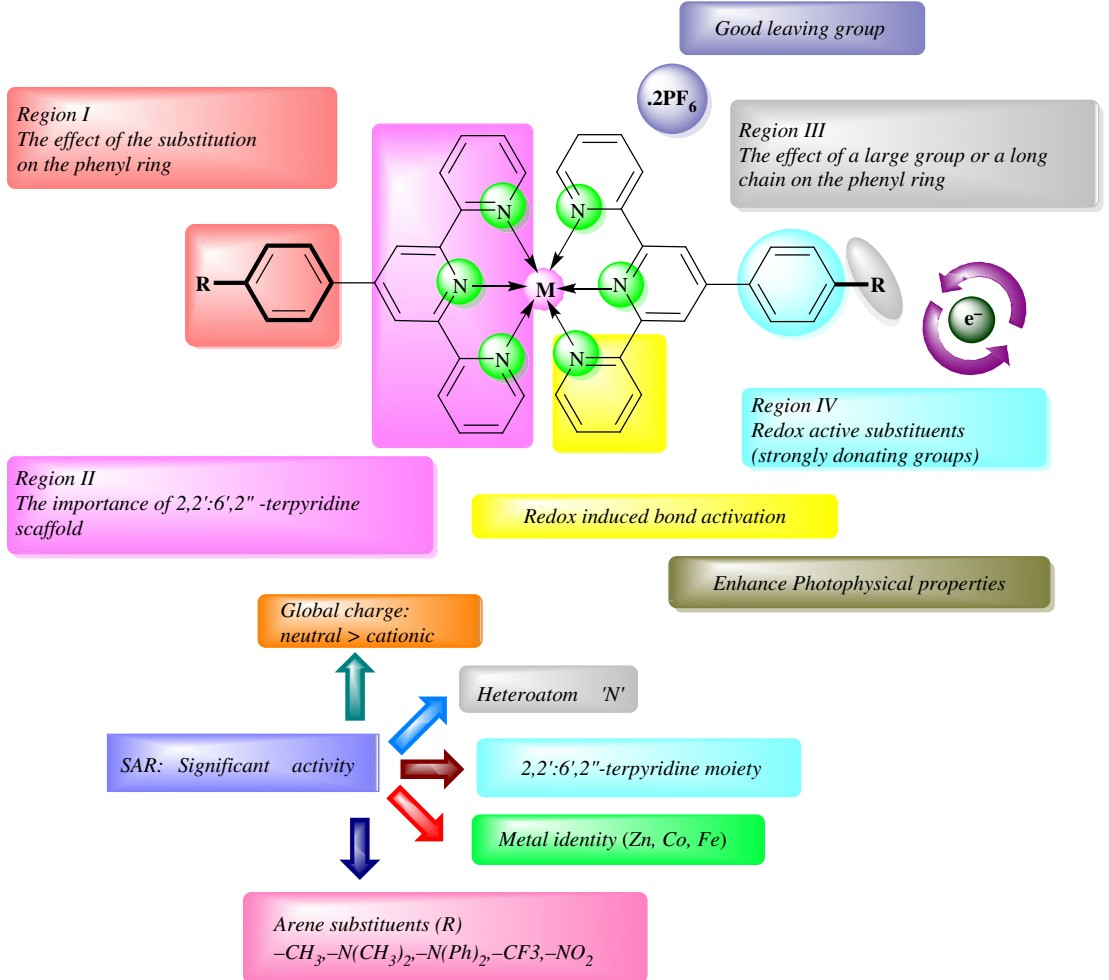

**Figure 5.** Structure–activity relationship of the synthesized terpyridine-based ligands and metal complexes.

within the entire chelate. In addition, the chelate cycle increases the liposolubility of central metal according to the concept of similarity and inter-miscibility, thereby promoting its permeation through the lipid layer of cell membranes and thus preventing enzyme activity resulting in excellent anti-bacterial properties.

## 3.44. Docking analysis

In the current study, molecular modelling studies were carried out using a molecular operating environment (MOE) program package for synthetic terpyridine derivatives ($L_1$–$L_9$) against shikimate dehydrogenase and penicillin-binding protein by targeting their active sites and examining the binding poses assumed by the ligands. Computational analysis of designed inhibitors was performed via docking simulations to evaluate the binding orientation, affinity and the binding energy of the tested inhibitors. Potential isolated binding sites of target shikimate dehydrogenase and penicillin-binding proteins were analysed by MOE software. The possible binding sites of ligand molecules at shikimate dehydrogenase are: Met1, Glu2, Thr3, Tyr4, Ile29, Glu30, His31, Pro32, Gly54, Gly55, Gly57, Glu74, Leu75, Thr76, Glu77, Arg78, Ala79, Ala80, Leu81, Ala82, Net89, Leu91, Leu97, Asp99, Asn100, Thr101, Asp102, Gly103, Val104, Leu107, Arg132, Gly133, Val134, Leu135, Leu136, Pro137, Leu139, Ser140, Leu141, Asp142, Phe162, His164, Thr165, His247, Leu250, Leu251, Gly254 (figure 6a); and at penicillin-binding proteins are: Arg408, Leu413, Asn417, Arg418, Phe421, Gly422, Pro424, Thr425, Leu437, Pro438, Asp440, Arg463, Lys470, Net474, Lys496, Glu497, Pro497, Asn501, Thr665, Gly666, Glu694, Glu695, Val696, Pro697, Ala698, Asp698, Tyr700, Gly701, Trp702, Ala734, Asn735, Val896 (figure 6b).

The possible strength of ligand–protein interaction was calculated based on minimal binding energy and scoring efficiency. 'Grid point' was allocated to the energy of interaction between the compound and

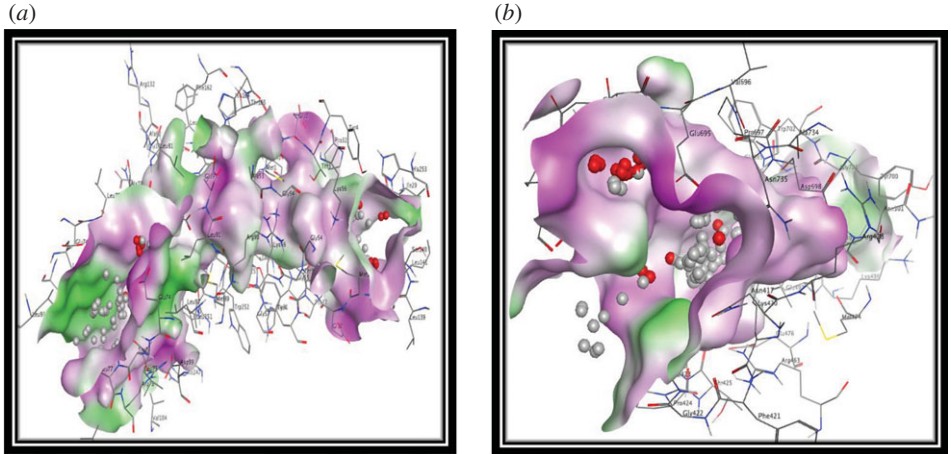

**Figure 6.** Three-dimensional docking pose of (*a*) shikimate dehydrogenase and (*b*) penicillin-binding protein having active binding sites.

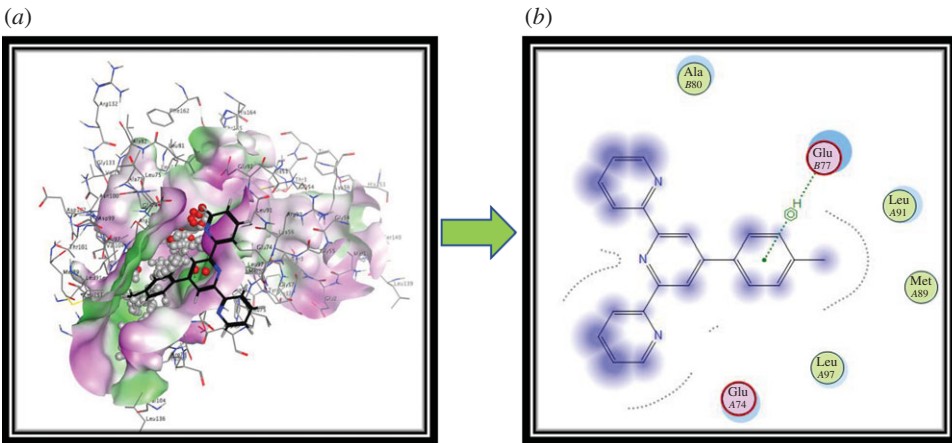

**Figure 7.** Molecular modelling views and information of ligand **L₁** association with atoms of shikimate dehydrogenase as the target (PDB ID: 3DON) performed by MOE software. (*a*) Three-dimensional interaction and (*b*) two-dimensional interaction with amino acid residues.

the target enzyme. Finally, those synthesized ligands were docked with the target molecules active sites. In short, at every simulation stage, the interaction energy of ligands and proteins was determined using atomic affinity potentials, calculated on a grid, while the remaining parameters were set as usual. The effects of molecular docking were clustered, and the lowest binding energy cluster was assessed as a binding representative state. The target proteins were considered successfully docked with ligand molecules as minimum binding energies were released. Each of the compounds has been checked to collect information on binding interactions that can be highly important for target inhibition. Binding interaction diagrams were obtained using the MOE ligand interaction analysis tool.

Among the L series, it was observed that against shikimate dehydrogenase (PDB ID: 3DON), the most active compound is ligand **L₁** ($-5.97$ kcal mol$^{-1}$) manifesting significant binding affinity. It develops the hydrophobic π–π stacked type associations with Glu77 and Glu74 of catalytic triad amino acid residues. Ala80, Leu91, Met89 and Leu87 amino acid residues of active pockets of shikimate dehydrogenase develop the hydrophobic π–π stacked, π-sulfur, hydrogen bond and hydrophobic π-alkyl type interactions, as shown in figure 7*a,b*.

Similarly, for penicillin-binding protein (PDB ID: 1VQQ), ligand **L₁** ($-5.743$ kcal mol$^{-1}$) showed the best activity (table 3). Additionally, the derivative **L₁** reveals its inhibitory potential against penicillin-binding protein by forming fruitful types of electrostatic interactions. This compound builds up hydrophobic π-alkyl type interactions with Glu695 of the acyl-binding pocket inside the penicillin-binding protein. Asn735 of the peripheral anionic site (PAS) develops hydrophobic π–π stacked type associations inside the pocket of penicillin-binding protein. This ligand also exhibits π-lone pair,

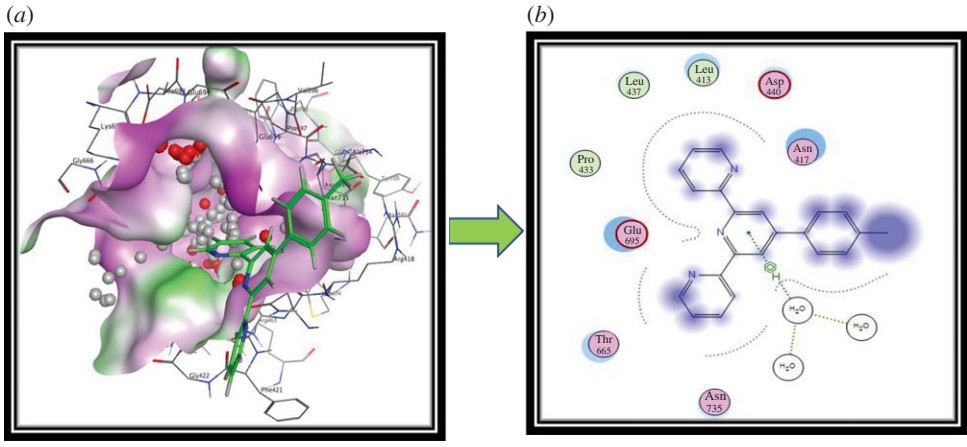

**Figure 8.** Molecular modelling views (best pose has been shown) and information of ligand **L₁** interaction with atoms of penicillin-binding protein as the target (PDB ID: 1VQQ) performed by MOE software. (*a*) Three-dimensional interaction and (*b*) two-dimensional interaction with amino acid residues.

**Table 3.** Energy values obtained by docking analysis of some of the synthesized ligands (**L₁–L₉**) against shikimate dehydrogenase and penicillin-binding protein target molecule.

| compound no. | shikimate dehydrogenase lowest binding energy $\Delta G$ (kcal mol$^{-1}$) | penicillin-binding protein lowest binding energy $\Delta G$ (kcal mol$^{-1}$) |
|---|---|---|
| **L₁** | −5.970 | −5.743 |
| **L₂** | −5.421 | −5.283 |
| **L₃** | −5.413 | −5.237 |
| **L₄** | −3.749 | −4.437 |
| **L₅** | −5.050 | −4.859 |
| **L₆** | −4.849 | −4.664 |
| **L₇** | −4.557 | −4.590 |
| **L₈** | −4.834 | −4.010 |
| **L₉** | −4.783 | −4.964 |
| standard | −5.010 (NSC) | −8.180 (gentamycin) |

hydrophobic π−π T-shaped and hydrophobic π-alkyl type associations with the Asn417, Asp440, Leu413, Leu437, Pro438 and Thr665 amino acid residues inside the active pockets of penicillin-binding protein as shown in figure 8*a,b*.

Overall, it has been observed that these ligands (**L₂–L₉**) showed moderate to good binding affinity with the target proteins. However, it is observed that **L₁** showed maximum binding affinities with shikimate dehydrogenase as compared with penicillin-binding proteins (figures 7 and 8). Geometric protein arrangement of amino acid residues inappropriate and disallowed regions shows the value of the target proteins.

## 3.45. Density functional theory (computational study)

### 3.45.1. Frontier molecular orbitals analysis

The energies and extents of the highest occupied molecular orbital (HOMO) and lowest unoccupied molecular orbital (LUMO) in a molecule are related to its ability to donate and accept an electron and the energy gap between these orbitals tells the stability, chemical reactivity and photoabsorption behaviour of the compounds [83]. Different functional groups in compounds can change the positions and energy gap of HOMO-LUMO, which noticeably affects the charge transfer transitions. This change in the energy band gap can be exploited in tuning photophysical and chemical reactivity features of the

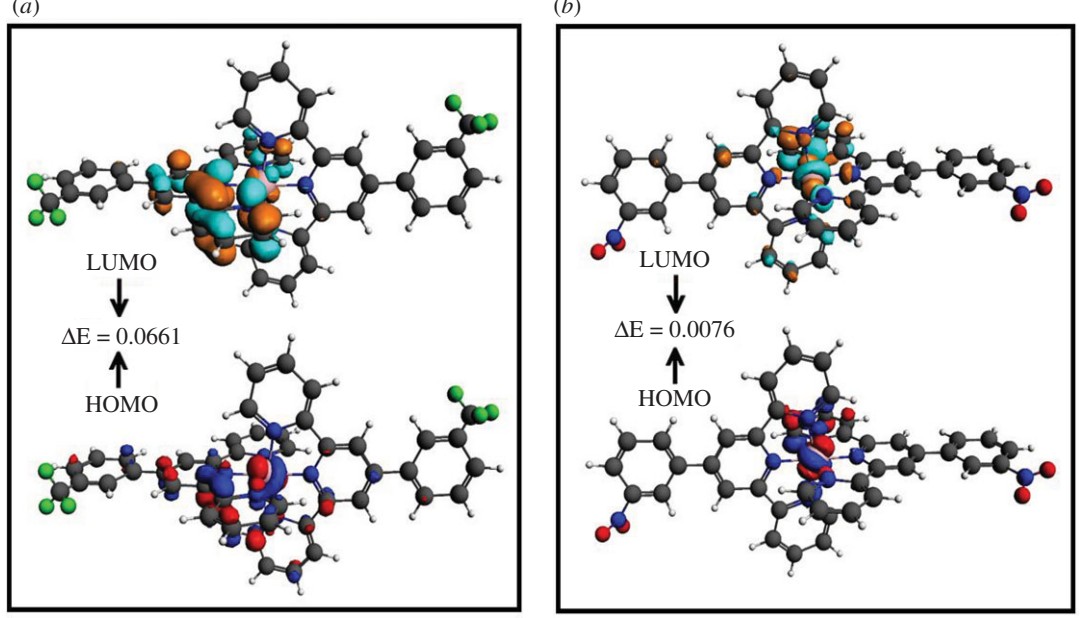

**Figure 9.** DFT-B3LYP* (DZ) calculated frontier molecular orbitals (FMOs) of (*a*) **C$_{15}$** and (*b*) **C$_{24}$**.

**Table 4.** DFT-B3LYP* (DZ) calculated values of $E_{LUMO}$, $E_{HOMO}$ and $E_{HOMO}$-$E_{LUMO}$ gap ($\Delta E$) of representative complexes experimental $\lambda_{max}$.

| compound code | $E_{LUMO}$ (Hartee) | $E_{HOMO}$ (Hartee) | $E_{LUMO}$-$E_{HOMO}$ gap $\Delta E$ (Hartee) | $\lambda_{max}$ (nm) (experimental) |
|---|---|---|---|---|
| **C$_{13}$** | −0.1505 | −0.2121 | 0.0616 | 570 |
| **C$_{14}$** | −0.1088 | −0.1235 | 0.0147 | 526 |
| **C$_{15}$** | −0.2257 | −0.2918 | 0.0661 | 568 |
| **C$_{22}$** | −0.1709 | −0.2136 | 0.0427 | 580 |
| **C$_{23}$** | −0.1244 | −0.1253 | 0.0009 | 535 |
| **C$_{24}$** | −0.0881 | −0.0957 | 0.0076 | 575 |

compounds. DFT-B3LYP* (DZ) calculated values of $E_{LUMO}$, $E_{HOMO}$ and $E_{HOMO}$-$E_{LUMO}$ gap ($\Delta E$) of the metal complexes (**C$_{13}$**, **C$_{14}$**, **C$_{15}$**, **C$_{22}$**, **C$_{23}$**, and **C$_{24}$**) are given in table 4. Frontier molecular orbitals participating in electron transfer in the complexes are depicted in figure 9 and electronic supplementary material, whereas experimentally determined $\lambda_{max}$ for all the compounds are shown in table 4. $\Delta E$ of the **C$_{15}$** was found to be higher among all corresponding compounds whereas **C$_{23}$** showed the smallest energy gap. Hence, tuning the energy band gap by variating the functional group can cause variation in absorptive properties (explained further in UV-Vis analysis section).

## 3.46. UV-Vis analysis

UV-Vis spectra of representative complexes of both terpyridine ligands were calculated by DFT-B3LYP*(DZ) methods to analyse the electronic absorption patterns in the complexes. A comparison of experimental UV-Vis and calculated spectra of compounds **C$_{15}$** and **C$_{24}$** is given in figure 10, which indicates good agreement between the two for both complexes.

The experimental absorption spectrum showed the typical metal to ligand charge transfer (MLCT) transition of the Fe(II)-terpyridine system (**C$_{15}$**) at $\lambda_{max}$ 568 nm, while in the calculated spectrum, it was observed at 572 nm. An intense absorption band in the calculated spectrum of Fe(II) complex (**C$_{24}$**) was observed at 578 nm (experimental $\lambda_{max}$ = 575 nm) which is due to an MLCT transitions in the complex. A redshift in theoretical and experimental absorption spectrum is attributed to the low energy gap for **C$_{24}$** ($\Delta E$ = 0.0076 Hartee) as compared with **C$_{15}$** ($\Delta E$ = 0.0661 Hartee). Hence, bandgap tuning by different substituents can alter the absorption behaviours and photophysical characteristics.

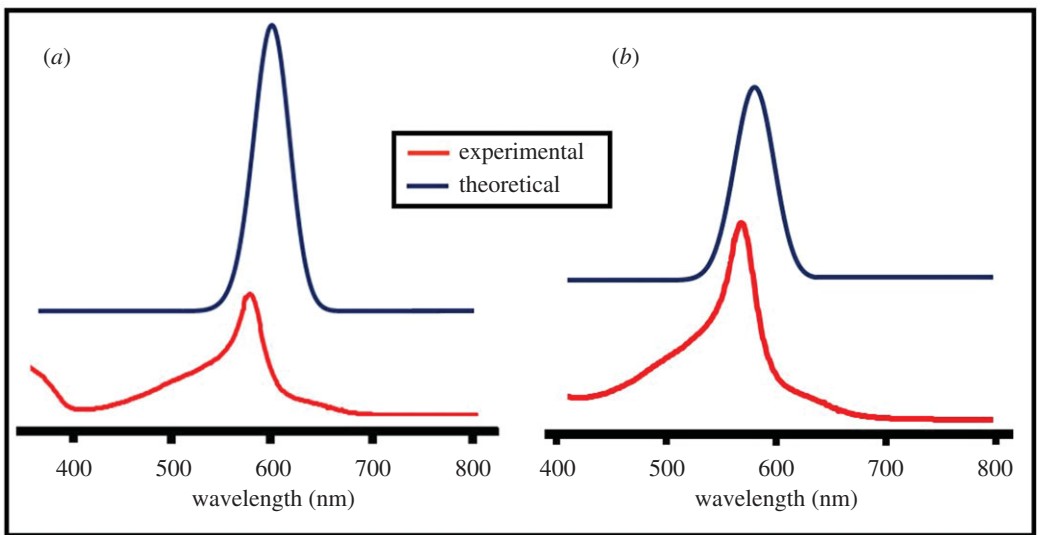

**Figure 10.** Experimental and DFT-B3LYP* (DZ) calculated UV-Vis spectra of representative metal complexes of ligands L$_5$ and L$_8$: (*a*) **C$_{15}$** and (*b*) **C$_{24}$**.

**Table 5.** The DFT-B3LYP* (DZ) calculated chemical reactivity parameters of representative complexes.

| compound code | electron affinity (EA) | ionization potential (IP) | electro-negativity (x) | electro-philicity (ω) | chemical potential (μ) | global hardness (η) | global softness (σ) |
|---|---|---|---|---|---|---|---|
| **C$_{13}$** | 0.1505 | 0.2121 | 0.1813 | 0.533599 | −0.1813 | 0.0308 | 16.23377 |
| **C$_{14}$** | 0.1088 | 0.1235 | 0.11615 | 0.917743 | −0.11615 | 0.00735 | 68.02721 |
| **C$_{15}$** | 0.2257 | 0.2918 | 0.25875 | 1.012883 | −0.25875 | 0.03305 | 15.12859 |
| **C$_{22}$** | 0.1709 | 0.2136 | 0.19225 | 0.865575 | −0.19225 | 0.02135 | 23.4192 |
| **C$_{23}$** | 0.1244 | 0.1253 | 0.12485 | 1.731947 | −0.12485 | 0.0045 | 111.111 |
| **C$_{24}$** | 0.0881 | 0.0957 | 0.0919 | 1.111264 | −0.0919 | 0.0038 | 131.5789 |

## 3.47. Chemical reactivity parameters

The chemical reactivity parameters indicate the capability of structures to be stabilized by attracting charge from the environment [84]. The chemical reactivity characteristics of the compounds can be evaluated by parameters such as ionization potential, electron affinity, electro-negativity, electro-philicity, chemical potential, global hardness and global softness [83] These parameters were calculated for the metal complexes (**C$_{13}$**, **C$_{14}$**, **C$_{15}$**, **C$_{22}$**, **C$_{23}$** and **C$_{24}$**) and are reported in table 5. The chemical reactivity characteristics electro-negativity ($x = IP + EA/2$) chemical hardness ($\eta = IP\text{-}EA/2$) and chemical potential ($\mu = E_{LUMO} + E_{HOMO}/2$) are calculated according to Koopmans Theorem [85] IP (-$E_{HOMO}$) is the ionization potential and EA (-$E_{LUMO}$) is the electron affinity. Global softness ($\sigma$) can be evaluated by the equation ($\sigma = 1/2\eta$) [86] The electro-philicity index ($\omega$) for measuring electrophilic strength was calculated by ($\omega = \mu^2/2\eta$) in accordance to Parr *et al.* [87,88].

A small energy gap indicates the compound to be soft and reactive, while a large energy gap indicates the molecule to be hard and not react easily [89]. The complexes **C$_{14}$**, **C$_{23}$** and **C$_{24}$** have shown greater softness character (68, 111 and 131, respectively) thereby tending to be more reactive, which is attributed to the lower bandgap values obtained. In addition, these compounds have low values of ionization potential indicating higher reactivities. Higher values of global hardness than softness indicate less reactivity and higher stability and from the calculated values of global hardness, the most thermally and kinetically stable compound is **C$_{22}$**. The calculated negative value of chemical potential shows that it is comparatively easy for the compounds to gain electrons from the environment. However, compounds are found to have greater charge transferability and most of the compounds are found to be reactive.

Strong charge transfers by FMO analysis, also indicated in the UV-Vis spectra of representative complexes, indicated that representative compounds could show potential activities against microbial strains. Tuning the bandgap by substituting different functional groups can alter reactivity characteristics along with photophysical properties as described in the UV-Vis analysis. It is evident from the chemical reactivity studies that these compounds can be used as efficient anti-microbial and biological agents for further studies.

# 4. Conclusion

We have presented the spectroscopic, photophysical, and anti-microbial studies of Zn(II), Co(II) and Fe(II) complexes of the variously substituted 2,2′:6′,2″-terpyridine motif. New substrates of symmetrically substituted $p$-aryl-2,2′:6′,2″-terpyridine have been efficiently prepared by employing a multi-step Kröhnke methodology. The resulting terpyridine-based complexes exhibit interesting photo-luminescent properties with strong emission upon excitation at the corresponding absorption maximum. Investigations carried on $[M(tpy-X)]^{2+}$ complexes (where X is the substituent in the 4′ position of 2,2′:6′,2″-terpyridine) have revealed that introduction of either electron-donating (-CH$_3$, -N(CH$_3$)$_2$, -N(Ph)$_2$, -OCH$_3$) or electron-withdrawing (-NO$_2$, -CF$_3$) groups onto the aryl-terpyridine scaffold alters absorbance as well as fluorescence properties and thus resulted in bathochromic shifted and structurally varied absorption and emission spectra. In addition, we have observed that the introduction of d-block transition metals (Zn, Co, Fe) into the terpyridine derivatives modified their absorption and emission properties as compared with terpyridine ligands (**L$_1$–L$_9$**), and most complexes (**C$_1$–C$_{27}$**) showed red-shifted absorption and emission spectra of varied appearance in contrast with the 2,2′:6″,2′-terpyridines. Many of the new compounds have interesting photo-luminescent properties with high emissions. Importantly, almost all the functionalized terpyridine complexes display longer emission $\lambda$ and they are stronger emissive in contrast with the unsubstituted 2,2′:6′,2″-terpyridine.

The ligands and complexes were also evaluated for their anti-microbial potential (*in vitro*). The analysis revealed that the complexes (**C$_1$–C$_{27}$**) exhibited more potent activity than free ligands (**L$_1$–L$_9$**). Overall, the entire compounds exhibited moderate to excellent anti-microbial activities. These results provide new possibilities for therapeutic purposes. The experimental results complemented with *in silico* studies provided insights into structure–property relationships. We are quite confident that the reported structures may find applications in organic electronics as well as medicines as potential materials and this will be the target of our future investigations. The investigation permitted the selection of materials with the most promising properties with special emphasis on the nature of the substituents.

Data accessibility. All the UV, fluorescence, mass spectra, biological evaluation images and DFT images of selected compounds can be found in the electronic supplementary material.
Authors' contributions. E.U.M. was involved in main idea, supervision and final writing the manuscript. M.M. was involved in co-supervision. A.S. was involved in co-supervision. S.F. was involved in experimental work performance and first-draft preparation. A.N. was involved experimental work performance and first-draft preparation. N.N. was involved in data analysis and collection, software. N.F. performed enzyme inhibition experiments. S.K. performed DFT studies. A.A.A. performed DFT studies. M.N.Z. performed molecular docking studies. B.A.K. was involved in mass spectrometry analysis. All authors gave final approval for publication.
Competing interests. There are no conflicts of interest to declare.
Funding. The Higher Education Commission of Pakistan (HEC) under project no. NRPU-6484 funded this study. M.M. gratefully acknowledges the financial support provided by the Ferdowsi University of Mashhad.
Acknowledgements. The authors are highly grateful to Dr. Renhao Dong, Department of Advanced Materials, Dresden University, Germany for his kind help in spectroscopic measurements.

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
