## [Reviewer comments · Royal Society Open Science]

Review History

RSOS-201208.R0 (Original submission)

Review form: Reviewer 1

Is the manuscript scientifically sound in its present form?

Yes

Are the interpretations and conclusions justified by the results?

Yes

Is the language acceptable?

Yes

Do you have any ethical concerns with this paper?

No

Have you any concerns about statistical analyses in this paper?

No

Recommendation?

Major revision is needed (please make suggestions in comments)

Comments to the Author(s)

Attached please find the comment details (Appendix A).

Review form: Reviewer 2**Is the manuscript scientifically sound in its present form?**

Yes

Are the interpretations and conclusions justified by the results?

Yes

Is the language acceptable?

Yes

Do you have any ethical concerns with this paper?

No

Have you any concerns about statistical analyses in this paper?

No

Recommendation?

Accept with minor revision (please list in comments)

Comments to the Author(s)

The authors provided complex research on terpyridines, their metal complexes and various properties of thereof. The obtained complex results demonstrate high level of novelty, and will be of interest to the broad range of scientists, even regarding the lack of X-Ray data for synthesized complexes.

However, some minor corrections are necessary prior to the publication:

1) The English in manuscript should be checked, some examples are:

p. 1 line 53 "metal complexes with the receptor proteins" – "corresponding proteins binding sites" (Name of the corresponding proteins should be added in to the abstract as well)

p. 2 line 58 "Constable 's" – eliminate extra spaces

p. 3 line 39 "whose" – which

p. 4 line 25 "than single ligand" - than a single ligand

p. 4 line 46 "theses" – these

p.5 line 18 "was added a substituted aryl aldehyde" – better a substituted aryl aldehyde was added, followed by

p. 5 line 25 "has removed" – was removed

p. 5 line 48 "pure complex" – a pure complex

p. 7 line 37 "a process of well diffusion" – the process of well diffusion

p. 7 line 39 "...experiments one Candida albicans" "...experiments: one Candida albicans" (missed colon)

p. 7 line 51 "Sabouraod" -Sabouraud

p. 8 line 55-56 "was shifted toward lower frequency" – "was shifted towards the lower frequency"

p. 26 line 18 "between ligands" - between the ligands

p. 29 line 32 "For instance, it was drawn impression" the meaning is unclear

2) Moreover, some phrases are recommended to be modified, such as:

“Well-known Kröhnke reaction” –better just “by Kröhnke reaction.”

3) The authors use the word «interesting» in the introduction, but it is unclear why they are interesting from the introduction itself. Thus, synonyms should be used, or an additional clarification of the term.

4) Appropriate citation for used docking software PDB is recommended.

5) Finally, regarding the fact that it might go beyond the proposed research, I would recommend the Authors to consider one small addition to their work, that might further increase the impact of their research. It would have been interesting to see a comparison of binding energies of ligands with any other known substance that binds with aforementioned proteins at the same site – to, let's say, put the data on the proposed substances in the overall context of bioactive molecules that target the corresponding sites of proteins. Thus, should the Authors consider it suitable, one additional docking experiment with a known substance that binds to the same site is recommended for inclusion.

Decision letter (RSOS-201208.R0)

Dear Professor Mirzaei:

Title: Terpyridine-metal complexes: Effects of different substituents on their physico-chemical properties and DFT studies
Manuscript ID: RSOS-201208

The editor assigned to your manuscript has now received comments from reviewers. We would like you to revise your paper in accordance with the referee and Subject Editor suggestions which can be found below (not including confidential reports to the Editor). Please note this decision does not guarantee eventual acceptance.

Please submit your revised paper before 08-Oct-2020. Please note that the revision deadline will expire at 00.00am on this date. If we do not hear from you within this time then it will be assumed that the paper has been withdrawn. In exceptional circumstances, extensions may be possible if agreed with the Editorial Office in advance. We do not allow multiple rounds of revision so we urge you to make every effort to fully address all of the comments at this stage. If deemed necessary by the Editors, your manuscript will be sent back to one or more of the original reviewers for assessment. If the original reviewers are not available we may invite new reviewers.

On behalf of the Subject Editor Professor Anthony Stace and the Associate Editor Dr Debashree Ghosh.

RSC Associate Editor:
Comments to the Author:
(There are no comments.)

RSC Subject Editor:
Comments to the Author:
(There are no comments.)

Reviewers' Comments to Author:
Reviewer: 1

Comments to the Author(s)
Attached please find the comment details.

Reviewer: 2

Comments to the Author(s)

The authors provided complex research on terpyridines, their metal complexes and various properties of thereof. The obtained complex results demonstrate high level of novelty, and will be of interest to the broad range of scientists, even regarding the lack of X-Ray data for synthesized complexes.

However, some minor corrections are necessary prior to the publication:

1) The English in manuscript should be checked, some examples are:

p. 1 line 53 "metal complexes with the receptor proteins" – "corresponding proteins binding sites" (Name of the corresponding proteins should be added in to the abstract as well)

p. 2 line 58 "Constable 's" – eliminate extra spaces

p. 3 line 39 "whose" – which

p. 4 line 25 "than single ligand" - than a single ligand

p. 4 line 46 "theses" – these

p.5 line 18 "was added a substituted aryl aldehyde" – better a substituted aryl aldehyde was added, followed by

p. 5 line 25 "has removed" – was removed

p. 5 line 48 "pure complex" – a pure complex

p. 7 line 37 "a process of well diffusion" – the process of well diffusion

p 7 line 39 "...experiments one Candida albicans" "...experiments: one Candida albicans"
(missed colon)

p. 7 line 51 "Sabourad" -Sabouraud

p. 8 line 55-56 "was shifted toward lower frequency" - "was shifted towards the lower frequency"

p. 26 line 18 "between ligands" - between the ligands

p. 29 line 32 "For instance, it was drawn impression" the meaning is unclear

2) Moreover, some phrases are recommended to be modified, such as:

"Well-known Kröhnke reaction" -better just "by Kröhnke reaction."

3) The authors use the word «interesting» in the introduction, but it is unclear why they are interesting from the introduction itself. Thus, synonyms should be used, or an additional clarification of the term.

4) Appropriate citation for used docking software PDB is recommended.

5) Finally, regarding the fact that it might go beyond the proposed research, I would recommend the Authors to consider one small addition to their work, that might further increase the impact of their research. It would have been interesting to see a comparison of binding energies of ligands with any other known substance that binds with aforementioned proteins at the same site - to, let's say, put the data on the proposed substances in the overall context of bioactive molecules that target the corresponding sites of proteins. Thus, should the Authors consider it suitable, one additional docking experiment with a known substance that binds to the same site is recommended for inclusion.

Author's Response to Decision Letter for (RSOS-201208.R0)

See Appendix B.

Decision letter (RSOS-201208.R1)

Dear Professor Mirzaei:

Title: Terpyridine-metal complexes: Effects of different substituents on their physico-chemical properties and DFT studies

Manuscript ID: RSOS-201208.R1

It is a pleasure to accept your manuscript in its current form for publication in Royal Society Open Science. The chemistry content of Royal Society Open Science is published in collaboration with the Royal Society of Chemistry.

On behalf of the Subject Editor Professor Anthony Stace and the Associate Editor Dr Debashree Ghosh.

RSC Associate Editor

Comments to the Author:

The authors have made the corrections suggested by the referees and therefore, I will be happy to accept the manuscript in its current state.

Reviewer(s)' Comments to Author:

Appendix A

Recommendation: Publish after major revisions noted.

In the present work, the spectroscopic, photophysical, antimicrobial and theoretical studies have been carried out to state the structure-property relationship of various species. The results obtained in this work may provide a valuable guidance for designing more efficient molecules for therapeutic purposes. However, some of their conclusions look weird. The authors should address the following items before considering acceptance.

Major comments:

1. On Page 23, the authors mentioned that “the absorption spectra of terpyridine-based metal complexes are very similar because all these compounds possess the same 2,2':6'2'' terpyridine central core”. This conclusion is very confusing. Because no similarity can be seen from the spectra shown in Table S1. For example, how can they say there is a similarity between the absorption spectrum of C_{11} and that of C_{27} ? Even if we only check the spectra of compounds with the same ligand but with different metal center (e.g., C_{10} , C_{11} and C_{12}), where is the similarity among them? More proper descriptions should be given.
2. On Pages 36-37, the authors give a conclusion that there is a good agreement between the calculated energy gap (ΔE) and the experimentally determined λ_{\max} . They also mentioned that lowering the energy gap causes and increase in λ_{\max} . However, after inspecting the values shown in Table 4, no agreement can be seen. For example, the ΔE of C_{15} is the largest one (0.0661), but its λ_{\max} is not the smallest one. So I have to say that there is an inconsistency between the description in the text and the data in the table. In addition, the ΔE of C_{23} is calculated wrong. The authors need to recheck their data very carefully and give proper conclusion.

Minor comments:

- 1) Why are the structures of C_{20} and C_{26} totally same in Scheme 1?
- 2) The symbol used to represent the same species should be unified, for example, “ π ” vs “pi”, “UV-Vis” vs “UV-Visible” and sometimes “ λ ” is in Italic but sometimes it is not.
- 3) On page 23, in line 14, the description of “...their metal complexes recorded in...” should be changed into “...their metal complexes are recorded in ...”.
- 4) On page 24, it should be “ d^{10} ” not “ d_{10} ” in line 46.
- 5) On page 24, a comma is needed between “complexes” and “286 nm” in line 55.
- 6) On page 25, the “p” of “p-substituted” in line 21 should be in Italic.
- 7) On page 30, it should be “spectrum of complex C_{13} ” in line 9.

Appendix B

Dated: 17 September 2020

Dr Laura Smith
Publishing Editor
Royal Society Open Science

(On behalf of the Subject Editor Professor Anthony Stace and the Associate Editor Dr Debashree Ghosh)

Subject: Reply to the Reviewers Comments

Thank you very much for your concern in publishing our manuscript titled “*Terpyridine-metal complexes: Effects of different substituents on their physico-chemical properties and DFT studies*” in your esteemed Journal. The envisioned manuscript has been edited according to the comments/suggestions of all referees. The revisions have been highlighted with “YELLOW COLOR”.

Please see below the amendments made as per their pieces of advice:

Reviewer 1

No.	Comments/Suggestions	Responses
1	Major Concerns: On Page 23, the authors mentioned that “the absorption spectra of terpyridine-based metal complexes are very similar because all these compounds possess the same 2,2':6'2” terpyridine central core”. This conclusion is very confusing. Because no similarity can be seen from the spectra shown in Table S1. For example, how can they say there is a similarity between the absorption spectrum of C₁₁ and that of C₂₇ ? Even if we only check the spectra of compounds with the same ligand but with different metal center (e.g., C₁₀ , C₁₁ and C₁₂), where is the similarity among them? More proper descriptions should be given.	We are agreed with learned Reviewer. Actually, there was a typographical mistake. Therefore, the ambiguous sentence has been omitted accordingly in the revised version.
2	On Pages 36-37, the authors give a conclusion that there is a good agreement between the calculated energy gap (ΔE)	Corrected and reformulated accordingly. We have revised and rephrased the discussion as per your recommendation to

	and the experimentally determined λ_{\max}. They also mentioned that lowering the energy gap causes an increase in λ_{\max}. However, after inspecting the values shown in Table 4, no agreement can be seen. For example, the ΔE of C₁₅ is the largest one (0.0661), but its λ_{\max} is not the smallest one. So I have to say that there is an inconsistency between the description in the text and the data in the table. In addition, the ΔE of C₂₃ is calculated wrong. The authors need to recheck their data very carefully and give proper conclusion.	remove any kind of inconsistency between the in text description and data in table. We have rechecked and recalculated the energy gap value of C₂₃ and corrected it in the Table 4.
1	Minor Issues: Why are the structures of C₂₀ and C₂₆ totally same in Scheme 1?	The typographical mistakes have been corrected. The concerned structures have been revised and modified accordingly.
2	The symbol used to represent the same species should be unified, for example, “π” vs “π”, “UV-Vis” vs “UV-Visible” and sometimes “λ” is in Italic but sometimes it is not.	All concerns have been addressed accordingly. Please see the changes in the revised manuscript.
3	On page 23, in line 14, the description of “...their metal complexes recorded in...” should be changed into “...their metal complexes are recorded in ...”.	Corrected and reformulated accordingly.
4	On page 24, it should be “d^{10}” not “d_{10}” in line 46.	Edited accordingly.
5	On page 24, a comma is needed between “complexes” and “286 nm” in line 55.	Corrected accordingly.
6	On page 25, the “p” of “p-substituted” in line 21 should be in Italic.	Corrected accordingly.
7	On page 30, it should be “spectrum of complex C13” in line 9.	Corrected accordingly.

Reviewer 2

No.	Comments/Suggestions	Responses
1	The authors provided complex research on terpyridines, their metal complexes and various properties of thereof. The obtained complex results demonstrate high level of novelty, and will be of interest to the broad range of scientists, even regarding the lack of X-Ray data for synthesized complexes.	Thanks for your kind appreciation!
1	Minor Issues: The English in manuscript should be checked, some examples are: p. 1 line 53 “metal complexes with the receptor proteins” –“corresponding proteins binding sites. (Name of the corresponding proteins should be added in to the abstract as well) p. 2 line 58 “Constable 's” – eliminate extra spaces p. 3 line 39 “whose” – which p. 4 line 25 “than single ligand”- than a single ligand p. 4 line 46 “theses” – these p.5 line 18 “was added a substituted aryl aldehyde” – better a substituted aryl aldehyde was added, followed by ... p. 5 line 25 “has removed” – was removed p. 5 line 48 “pure complex” – a pure complex p. 7 line 37 “a process of well diffusion” – the process of well diffusion p 7 line 39 “...experiments one Candida albicans” “...experiments: one Candida albicans” (missed colon) p. 7 line 51 “Sabouraod” –Sabouraud	The whole manuscript has been revised almost and corrected, wherever needed. To the best of our knowledge, the typographical and grammatical mistakes have been corrected. We apologize in advance for any overlooked error.

	p. 8 line 55-56 “was shifted toward lower frequency” – “was shifted towards the lower frequency” p. 26 line 18 “between ligands” - between the ligands p. 29 line 32 “For instance, it was drawn impression” the meaning is unclear	
2	Moreover, some phrases are recommended to be modified, such as: “Well-known Kröhnke reaction” –better just “by Kröhnke reaction.”	Corrected and modified accordingly.
3	The authors use the word «interesting» in the introduction, but it is unclear why they are interesting from the introduction itself. Thus, synonyms should be used, or an additional clarification of the term.	Corrected and reformulated accordingly.
4	Appropriate citation for used docking software PDB is recommended.	The appropriate citation has been mentioned accordingly. Please see the changes in revised manuscript.
5	Finally, regarding the fact that it might go beyond the proposed research, I would recommend the Authors to consider one small addition to their work, that might further increase the impact of their research. It would have been interesting to see a comparison of binding energies of ligands with any other known substance that binds with aforementioned proteins at the same site – to, let’s say, put the data on the proposed substances in the overall context of bioactive molecules that target the corresponding sites of proteins. Thus, should the Authors consider it suitable, one additional docking experiment with a known substance that binds to the same site is recommended for inclusion.	Thank you very much for your kind suggestion. It is added to your kind information that we have already analyzed and published such type of interactions of other ligands with the same proteins. Remarkably, the binding energies reported for terpyridine-based ligands/complexes are lower than that of already reported results. This makes this article superior to the previous publications. Could you please have a look over the following publications? 1) Ashraf, J., et al. (2017). "Design, Synthesis and Antibacterial Activities

		of New Azo-compounds: An Experimental and a Computational Approach." Letters in Drug Design & Discovery 14(10): 1145-1154. 2) Mughal, E. U., et al. (2018). "Design, synthesis and biological evaluation of novel dihydropyrimidine-2-thione derivatives as potent antimicrobial agents: experimental and molecular docking approach." Letters in Drug Design & Discovery 15(11): 1189-1201.
--	--	--

Please find enclosed the revised manuscript for your kind consideration.
Thank you very much in advance for your understandings and ongoing acceptance.

I look forward to hearing from you.
Best regards,

Prof. Dr. Masoud Mirzaei

P.S: Once again we apologize in advance for any overlooked mistake. Nonetheless, we have tried our best to improve the quality of this manuscript for your readers.